# Autonomous AI Assistant for Semiconductor Electron Micrograph Analysis: Instruction-Tuning Small-Scale Language-and-Vision Assistant for Enterprise Adoption in Low-Resource Settings

## Abstract

Semiconductor imaging and analysis are critical yet understudied in deep learning, limiting our ability for precise control and optimization in semiconductor manufacturing. We introduce a small-scale multimodal framework for analyzing semiconductor electron microscopy images (`MAEMI`) through vision-language instruction tuning. We generate a customized instruction-following dataset using large multimodal models on microscopic image analysis. We perform knowledge transfer from larger to smaller models through knowledge distillation, resulting in improved accuracy of smaller models on visual question answering (VQA) tasks. This approach eliminates the need for expensive, human expert-annotated datasets for microscopic image analysis tasks. `MAEMI`, can assist, accelerate, and even automate the semiconductor electron microscopy image ananysis tasks. Enterprises can further fine-tune `MAEMI` on their intellectual data, enhancing privacy and performance on low-cost consumer hardware. Our experiments show that `MAEMI` outperforms traditional methods, adapts to data distribution shifts, and supports high-throughput screening.

## 1 Introduction

Semiconductors, crucial for modern electronics:, undergo a complex multi-step production process. Fabless firms such as Qualcomm and NVIDIA design and simulate chip functionalities, while manufacturing is outsourced to foundries like TSMC and Samsung. Foundries handle semiconductor chip fabrication, which includes photolithography to imprint circuit patterns on silicon wafers, etching and doping for circuit formation, and intricate layering for circuit interconnection. After fabrication, chips undergo quality assurance, including electrical and stress testing, to confirm performance and defect-free status. Packaged semiconductors are assembled into devices like microprocessors and memory chips, integrated into various electronic systems, such as consumer electronics, automotive technologies, and space applications. Miniaturization is crucial to the semiconductor industry, enabling the creation of smaller, more powerful, and more efficient devices that advance the capabilities and functionality of electronic products. However, this pursuit faces challenges that require precision and control to ensure system-level performance and overcoming manufacturing inaccuracies. To tackle these obstacles, the industry leverages sophisticated imaging techniques for thorough testing and analysis. The relentless pursuit of miniaturization in semiconductor manufacturing demands an ever-increasing focus on achieving nanoscale precision. Advanced tools, such as scanning electron microscopy (SEM) and transmission electron microscopy (TEM), play a vital role in the semiconductor industry's push for precision. These electron beam instruments offer high-resolution micrographs (microscopic images), revealing intricate details of semiconductor materials and structures at the nanoscale. Their sophisticated imaging capabilities are crucial for quality control, including failure analysis, allowing precise characterization of microstructures. As indispensable assets in ensuring semiconductors conform to design specifications, these tools help enable subsequent process optimization or design adjustments to mitigate defects. Characterizing materials at the nanoscale is critical to driving ongoing technological progress. However, current technology falls short in effectively addressing the full spectrum of complexities and specialized requirements for material characterization in the semiconductor industry, particularly in accurate labeling and analysis of electron micrographs. Therefore, recent advancements in Artificial Intelligence (AI), including Large Multimodal Models (LMMs) like Gemini[Team *et al.*, 2023] and GPT-4 Turbo with Vision[OpenAI, 2023], which combine advanced natural language processing with visual understanding capabilities, can significantly impact the semiconductor manufacturing process in several ways. These vision-language models allow for the analysis of high-resolution electron micrographs, revealing intricate nanoscale structures of semiconductor materials. By identifying and providing insights into patterns, the multimodal large language models enable quality control and improve the precision and efficiency of semiconductor manufacturing. While proprietary, general-purpose LMMs offer benefits, their adoption faces challenges due to concerns regarding sharing enterprise data. Sharing sensitive information with third-party services could expose novel designs and processes, undermining semiconductor firms' intellectual property portfolio and jeopardizing future innovation. Conversely, open-source, small-scale multimodal models (SMMs) like LLaVA[Liu *et al.*, 2023] and MiniGPT-4[Zhu *et al.*, 2023] can be more cost-effective for task-specific customization on mi-

croscopic image analysis, enabling safe, reliable, on-premises enterprise adoption. The smaller multimodal models offer better interpretability due to their open-source nature. However, they may not match the reasoning and generalization capabilities of proprietary LMMs, sometimes producing less coherent and contextually relevant outputs. In addition, generating high-quality training datasets is crucial for customizing SMMs for microscopic image analysis, but acquiring such datasets is scarce and expensive. The annotation process requires expert knowledge and specialized tools, making it time-consuming and resource-intensive. Additionally, the diverse image characteristics and representations resulting from the different imaging techniques pose a significant challenge to developing a generalizable multimodal model that can perform effectively across various electron micrograph-based datasets. Furthermore, electron micrograph-based image-captioning and open-ended VQA tasks are promising but challenging due to complex image characteristics, such as high intra-class dissimilarity, high inter-class similarity, and spatial heterogeneity (refer Figure 1). These complexities pose obstacles to accurate image understanding and question answering.

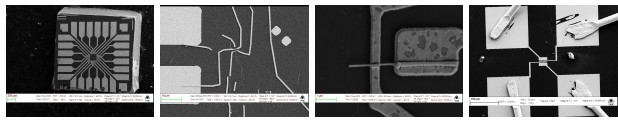

(a) High intra-dissimilarity in electron micrographs of MEMS devices.

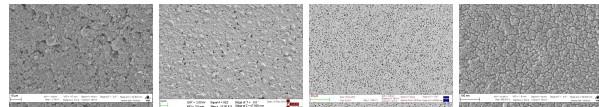

(b) High inter-class similarity in electron micrographs of various nanomaterials: powders, films, porous structures, and particles.

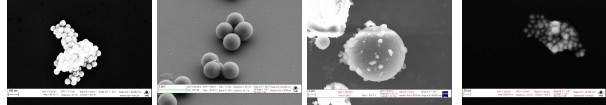

(c) Spatial heterogeneity of uneven size distribution in nanoparticle micrographs.

Figure 1: Challenges in analyzing electron micrographs from the SEM dataset.

To address the challenges of privacy concerns, scarcity of high-quality data, and small-scale models generalization and interpretability, our study introduces a novel approach called 'On-Premises Secure Multimodal Instruction Tuning of SMMs'. This approach enables SMMs to achieve performance comparable to larger models through transfer learning, all while decreasing computational requirements. It follows a 'teaching-via-data' method and utilizes state-of-the-art, vision-language models to generate custom instruction-following data on niche tasks to train smaller models for task-specific adaptation, avoiding the need for human-annotated data. Our approach empowers enterprises to fine-tune small-scale, pre-trained multimodal models on their own data within their infrastructure, enhancing privacy, security, and reducing computational costs, while improving their ability to respond to complex multimodal inputs. Overall, it offers a promising solution to the limitations of existing proprietary LMMs, potentially

democratizing access to their high-end capabilities and accelerating their adoption across a wide range of tasks. To address the challenges of privacy concerns, scarcity of high-quality data, and small-scale models generalization and interpretability, our study introduces a novel approach called 'On-Premises Secure Multimodal Instruction Tuning of SMMs'. This approach enables SMMs to achieve performance comparable to larger models through transfer learning, while decreasing computational requirements. It follows a 'teaching-via-data' method and utilizes state-of-the-art, vision-language models to generate custom instruction-following data on niche tasks. This synthetic data is used to train smaller models for task-specific customization, avoiding the need for human-annotated data. Our approach empowers enterprises to fine-tune smaller, pre-trained models on their own data within their infrastructure, enhancing privacy, security, and reducing computational costs, while improving their ability to respond to complex multimodal inputs. Overall, it offers a promising solution to the limitations of existing proprietary LMMs, potentially democratizing access to their high-end capabilities and accelerating their adoption across a wide range of tasks. In this work, we present the Multimodal Assistant for Electron Micrograph Analysis (MAEMI), an end-to-end trained, small-scale multimodal model designed for microscopic image analysis. We utilize visual-language instruction tuning to customize MAEMI on microscopic image analysis using GPT-4-Turbo with Vision generated high-fidelity multimodal labeled data, eliminating the need for additional human annotation efforts. The generated instruction-following dataset comprises image-question-answer pairs that delve into various aspects of nanomaterials in microscopic images, created by prompting a large-scale, pre-trained multimodal model (like GPT-4 Turbo with Vision) with task-specific instructions based on the target microscopic images. The high-quality generated dataset trains the proposed framework to analyze electron microscopy images of nanomaterials, enabling it to answer questions about the content within the visual inputs. Our approach empowers smaller models with zero-shot learning capabilities, enabling them to grasp both the intricate context within microscopic images, including spatial relationships and object interactions, and the nuanced semantics and intent behind the questions. Consequently, this leads to improved grounded language generation and visual reasoning capabilities, resulting in more accurate answers. Furthermore, our approach facilitates knowledge distillation from larger to smaller models, ultimately enhancing their performance to be on par with larger models in microscopic image analysis tasks. Our novel encoder-decoder multimodal framework efficiently processes and aligns images and text to generate textual responses to questions across image captioning and open-ended VQA tasks. Key components of MAEMI for the zero-shot image captioning task are illustrated in Figure 2. The multimodal model, MAEMI, integrates visual processing and language modeling for answering questions about specific image features. It includes: (a) The vision encoder, using a vision transformer[Dosovitskiy *et al.*, 2020], analyzes the microscopic images by splitting them into patches and using self-attention mechanism to capture beyond pair-wise patch relationships. This allows for understanding the global context and highlighting relevant visual regions and

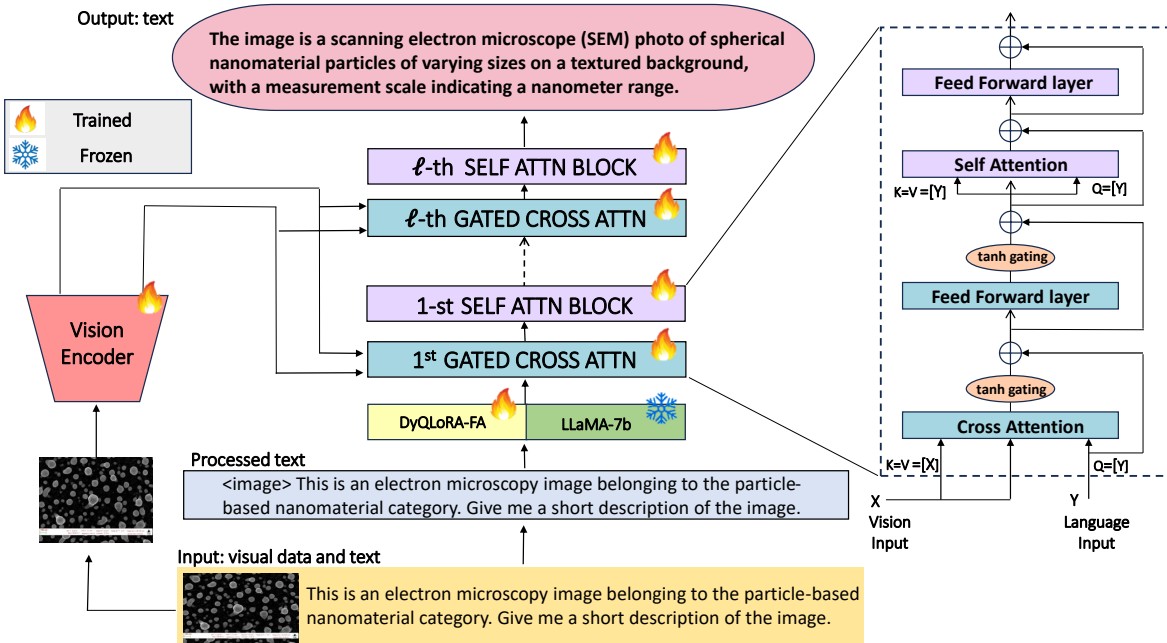

Figure 2: The schematic illustrates MAEMI, a small-scale, autoregressive text generation model. It takes as input a multimodal prompt consisting of the target image interleaved with auxiliary image descriptions and captioning instructions (or end-user questions), and outputs visually grounded descriptive text in a zero-shot setting. MAEMI utilizes a vision transformer and a pre-trained language model to analyze images and interpret questions about them. Both encoders synergize through a multi-layer structure of alternating gated cross-attention and self-attention blocks, effectively integrating both modalities – visual and textual information – to generate accurate and contextually relevant answers. The framework is trained in a supervised learning setting using a vision-language instruction tuning dataset to generate answers that are grounded in visual information and aligned with the target image content.

relationships. A $<cls>$ token attends to and aggregates information from all patches, resulting in a higher-level visual semantic representation to capture the overall context or summary of the input image. (b)The text encoder, crucial for analyzing end-user questions, takes as input an interleaved multimodal prompt. We insert $<image>$ token in the prompt at the image location and append an $<Encode>$ token to facilitate multimodal integration, with its output embedding symbolizing the fused representation. The text encoder leverages instruction-tuned Llama-2-7b, a pretrained language model, to capture language nuances and context. The language-only model is customized using parameter-efficient fine tuning technique, enhancing its ability to interpret end-user questions. Both the vision and language-only unimodal encoders synergize to interpret end-user questions and analyze visual input for generating answers consistent with the visual context. (c) It utilizes a multi-layered structure with multiple blocks, alternating between self-attention and gated cross-attention blocks. This design facilitates complex interactions between visual and textual modalities. By extracting and refining information from both modalities at each level, the framework progressively builds a comprehensive understanding, enabling coherent and contextually relevant answers to the end-user questions. Gated cross-attention blocks integrate visual features with textual features. The gating mechanism acts as a non-linear filter and controls the flow of information from the vision encoder to the language processing cross-attention

blocks, allowing the framework to focus on relevant visual features for the text generation task. Self-attention blocks, on the other hand, allow the framework to weigh the importance of different parts of the fused information. Within the self-attention blocks, this is used to refine the text features based on their context within the text itself. We train the framework in a supervised learning setting, minimizing language modeling loss to ground its text generation in visual information. This results in accurate answers closely aligned with the image content, empowering the framework with microscopic image analysis expertise. In summary, the proposed framework, trained through vision-language instruction tuning, takes as input a multimodal prompt of microscopic images paired with auxiliary image descriptions, and outputs free-form text as answers to a range of open-ended, image-related questions.

## 1.1 Dynamic Low-Rank Adaptation with Activation Memory Reduction (DyQLoRA-FA)

Low-Rank Adaptation (LoRA[Hu *et al.*, 2021]) is a deep learning technique used to efficiently fine-tune large-scale pre-trained language models on consumer hardware to adapt for niche domain-specific tasks. It accomplishes this without introducing additional inference latency and without the need for extensive retraining. LoRA adapts these large-scale models to domain-specific tasks by preserving the vast knowledge acquired during pretraining, thereby avoiding catastrophic forgetting—a phenomenon where the language model loses previously learned information while acquiring new infor-

mation. This selective adaptation of large pre-trained language models is achieved by inserting small pairs of trainable low-rank weight matrices, known as adapters, into each pre-trained model layer. By keeping the original pretrained model weights unchanged, LoRA updates only these auxiliary parameters, achieving comparable performance to full-parameter fine-tuning. LoRA primarily focuses on the linear layers in Transformer-based large-scale language models [Vaswani *et al.*, 2017], for several key reasons: (a) These layers are prevalent in such architectures and contain a significant portion of the language model's parameters. (b) They are well-suited for low-rank approximations, offering a balance between language model adaptability and computational efficiency. (c) Additionally, modifying linear layers directly impacts the language model's learning capabilities, making them ideal targets for efficient and effective fine-tuning. By taking advantage of the distinct features of linear layers, LoRA incorporates additional trainable parameters ($\Delta \mathbf{W}$) to capture task-specific information, thereby updating the pretrained language model without altering the original weights ($\mathbf{W}_0$). The low-rank adaptation, in which the original weight matrices are transformed by adding the product of pair of low-rank matrices, effectively allows the pretrained language model to learn domain-specific tasks, as expressed below:

$$\mathbf{Y} = (\mathbf{W}_0 + \Delta \mathbf{W})\mathbf{X} = \mathbf{W}_0\mathbf{X} + (\alpha \mathbf{A}\mathbf{B})\mathbf{X} \qquad (1)$$

Here, $\mathbf{Y} \in \mathbb{R}^{b \times d_{\text{out}}}$ and $\mathbf{X} \in \mathbb{R}^{b \times d_{\text{in}}}$ represent the output and input tensors, respectively. We omit the bias term for simplicity. $d_{\text{in}}$ and $d_{\text{out}}$ denote the input and output dimensions, respectively. $b$ denotes the batch size. The original weight matrix, denoted as $\mathbf{W}_0 \in \mathbb{R}^{d_{\text{in}} \times d_{\text{out}}}$, preserves the pretrained knowledge. $\Delta \mathbf{W}$, the low-rank approximation added to $\mathbf{W}_0$ during language model adaptation, enables fine-tuning for domain-specific tasks while preserving general capabilities. The projection-down weight matrix $\mathbf{A}$ has dimensions $\mathbb{R}^{d_{\text{in}} \times r}$, and the projection-up weight matrix $\mathbf{B}$ has dimensions $\mathbb{R}^{r \times d_{\text{out}}}$. The rank of the decomposition, denoted as $r$, is significantly smaller than $d_{\text{in}}$ or $d_{\text{out}}$ (i.e., $r \ll d_{\text{in}}$ or $d_{\text{out}}$). $\alpha$, a positive constant, is typically valued at $\frac{1}{r}$. The rank, $r$, is a critical hyperparameter that influences the balance between the pretrained language model's adaptation capacity, computational efficiency, and overall performance during the fine-tuning process for task-specific customization. During training, the low-rank weight matrices $\mathbf{B}$ and $\mathbf{A}$ are updated, while $\mathbf{W}_0$ remains fixed. During the fine-tuning of pre-trained language models, gradients for each trainable parameter are calculated using the loss function. These gradients guide optimizers, such as Adam[Kingma and Ba, 2014] or SGD[Robbins and Monro, 1951], in updating the trainable parameters. Additionally, optimizers maintain extra state information for these parameters, which includes momentum and adaptive learning rates. Thus, fine-tuning pre-trained language models necessitates storing not only the model parameters but also their gradients and optimizer states in memory. LoRA proportionally decreases the memory overhead associated with the gradients and optimizer states by reducing the number of trainable parameters through low-rank adaptation. This reduction is crucial for task-specific fine-tuning of large-scale language models. Consequently, LoRA requires fewer computational resources than full fine-tuning, making it a more efficient and scalable approach for adapting pre-trained language models to specific tasks. However, substantial memory is still necessary to store the large input activations (i.e., the high-dimensional intermediate outputs of layers, such as $\mathbf{X}$ in Equation 1) during the feed-forward pass. This is necessary for computing the gradients of the low-rank weights during backpropagation. High activation memory demands significantly limit scalability, especially when computational resources are constrained. Approaches such as selective LoRA [Hu *et al.*, 2021] or activation recomputation [Chen *et al.*, 2016] can potentially alleviate these demands, but suffer from trade-offs in terms of performance and efficiency. In conclusion, while LoRA enables efficient adaptation of pre-trained language models to specific tasks or domains, addressing the substantial activation memory demands during fine-tuning remains a key challenge. LoRA-FA [Zhang *et al.*, 2023] significantly reduces the activation memory footprint by freezing the pre-trained weights ($\mathbf{W}_0$), the projection-down weight ($\mathbf{A}$), and updating only the projection-up weight ($\mathbf{B}$) in each linear layer. In LoRA-FA, the frozen $\mathbf{A}$ is randomly initialized from a normal distribution, while $\mathbf{B}$ is initialized to zero and updated during fine-tuning. This approach allows for the computation of gradients solely for $\mathbf{B}$, leading to a substantial reduction in computational load. Moreover, it necessitates storing only the reduced-dimensionality input to $\mathbf{B}$ (i.e., $\mathbf{AX}$), where $\mathbf{A}$ maps the high-dimensional input $\mathbf{X}$ to a significantly smaller $r$-dimensional space, facilitating the computation of gradients for $\mathbf{B}$ during backpropagation with reduced activation memory. This approach significantly reduces the activation memory requirements without compromising fine-tuning performance and without introducing additional computational overhead and inference latency. Consequently, it enables efficient fine-tuning of pre-trained language models under resource constraints while preserving accuracy and minimizing memory consumption. However, LoRA-FA may have potential limitations, including potentially slower convergence rates in the initial stages of fine-tuning and the need for careful hyperparameter optimization of rank $r$ to achieve peak performance. Furthermore, LoRA-FA is a static low-rank adapter that works only with a specifically trained rank r. To address these limitations, DyLoRA[Valipour *et al.*, 2022] introduces dynamic low-rank adapters that are trainable and deployable across a range of ranks, thereby eliminating the need to find the optimal rank through multiple trainings. Dynamic low-rank adapters offer several key benefits. Firstly, their ability to dynamically adjust their rank allows for an optimal trade-off between computational efficiency and pre-trained language model performance on specialized domain-specific tasks. Secondly, because these adapters can adapt their rank according to the specific task and data distribution, they are particularly well-suited for scenarios involving continuous learning or frequent changes in data distributions, especially when facing out-of-distribution (OOD) data. We utilize DyLoRA to train and deploy LoRA-FA across a range of ranks, $r \in \text{Range}[r_{\min}, r_{\max}]$, with $r_{\min}$ and $r_{\max}$ as new hyperparameters. During training at each step, a rank $b$ is sampled from a pre-defined categorical distribution, $b \sim p_B(\text{Range}[r_{\min}, r_{\max}])$ and the matrices are truncated to $\mathbf{A}^{\downarrow b}$ and $\mathbf{B}^{\downarrow b}$ as follows:

$$\mathbf{B}^{\downarrow b} = \mathbf{B}[1:b,:]$$
$$\mathbf{A}^{\downarrow b} = \mathbf{A}[:,1:b]$$
$$\mathbf{Y} = \mathbf{W}_0\mathbf{X} + (\alpha\mathbf{A}^{\downarrow b}\mathbf{B}^{\downarrow b})\mathbf{X}$$

where $\mathbf{A}^{\downarrow b}$ and $\mathbf{B}^{\downarrow b}$ are the truncated forms of $\mathbf{A}$ and $\mathbf{B}$ at rank $b$, the back-propagation involves computing gradients $\frac{\partial\mathcal{L}}{\partial\mathbf{A}^{\downarrow b}}$ and $\frac{\partial\mathcal{L}}{\partial\mathbf{B}^{\downarrow b}}$, where $\mathcal{L}$ is the loss function. The back-propagation technique aims to update these matrices based on the loss function, taking into account the dynamic adaptation in rank. We compute gradient with respect to $\mathbf{B}$ as follows: Consider the contribution to the output $\mathbf{Y}$ from $\mathbf{B}$: $\mathbf{Y}_B = (\alpha\mathbf{A}^{\downarrow b}\mathbf{B}^{\downarrow b})\mathbf{X}$. The gradient of the loss $\mathcal{L}$ with respect to $\mathbf{B}^{\downarrow b}$ is:

$$\frac{\partial\mathcal{L}}{\partial\mathbf{B}^{\downarrow b}} = \frac{\partial\mathcal{L}}{\partial\mathbf{Y}_B} \cdot \frac{\partial\mathbf{Y}_B}{\partial\mathbf{B}^{\downarrow b}}$$
$$\frac{\partial\mathcal{L}}{\partial\mathbf{B}^{\downarrow b}} = \alpha\mathbf{A}^{\downarrow b}\left(\frac{\partial\mathcal{L}}{\partial\mathbf{Y}_B}\mathbf{X}\right)$$

Similarly, the gradient of the loss $\mathcal{L}$ with respect to $\mathbf{A}^{\downarrow b}$ is:

$$\frac{\partial\mathcal{L}}{\partial\mathbf{A}^{\downarrow b}} = \frac{\partial\mathcal{L}}{\partial\mathbf{Y}_B} \cdot \frac{\partial\mathbf{Y}_B}{\partial\mathbf{A}^{\downarrow b}}$$
$$\frac{\partial\mathcal{L}}{\partial\mathbf{A}^{\downarrow b}} = \alpha\mathbf{B}^{\downarrow b}\left(\frac{\partial\mathcal{L}}{\partial\mathbf{Y}_B}\mathbf{X}\right)$$

The gradients are used to update the trainable parameters using an optimizer like Adam or SGD as follows,

$$\mathbf{B}_{\text{new}}^{\downarrow b} = \mathbf{B}^{\downarrow b} - \eta \cdot \frac{\partial\mathcal{L}}{\partial\mathbf{B}^{\downarrow b}}; \mathbf{A}_{\text{new}}^{\downarrow b} = \mathbf{A}^{\downarrow b} - \eta \cdot \frac{\partial\mathcal{L}}{\partial\mathbf{A}^{\downarrow b}}$$
$$\mathbf{B}[1:b,:] = \mathbf{B}_{\text{new}}^{\downarrow b}; \mathbf{A}[:,1:b] = \mathbf{A}_{\text{new}}^{\downarrow b}$$

where $\eta$ is the learning rate. We manage the computational complexity associated with varying ranks in DyLoRA-FA through custom gradient accumulation and rank normalization. Gradient accumulation enables more stable and efficient learning by collecting gradients over multiple iterations, while rank normalization equalizes the impact of different ranks on language model fine-tuning by scaling gradients according to rank size. We employ weight-only quantization (WOQ) for fine-tuning pre-trained language models. WOQ compresses the original weights of the pre-trained language model by converting its high-precision weights (usually 16-bit floating-point) into lower-precision formats (e.g., 8-bit integers). This results in a drastic reduction in the language model's memory footprint and computational requirements. We fine-tune the quantized pre-trained language model on specific datasets related to the target domain-specific task using the parameter-efficient fine-tuning (PEFT) technique such as DyLoRA-FA, which compensates for any accuracy drops resulting from quantization. DyQLoRA-FA, which involves quantization, has been found to reduce memory requirements significantly, albeit at the cost of a slightly longer training time. This trade-off is generally considered acceptable, especially when it allows for the use of low-cost GPUs. In summary, DyQLoRA-FA is a flexible and efficient method for fine-tuning large language models across various rank sizes. It maintains performance without retraining, is highly memory-efficient, has low computational cost, and achieves comparable performance to full-parameter fine-tuning on diverse tasks.

## 1.2 Fine-Tuning, Pretrained Large Language Models(LLMs)

Llama 2[Touvron *et al.*, 2023], an advanced autoregressive pretrained language transformer built for natural language processing (NLP) tasks, leverages supervised fine-tuning (SFT) and reinforcement learning with human feedback (RLHF) to generate responses ideal for chat applications and various language generation tasks. Its robust foundation in understanding and generating human-like text, combined with its ability to effectively interpret and produce natural language, makes it well-suited for complex NLP tasks. Llama-2's architecture comprises 32 layers and 32 attention heads, efficiently handling large token sequences of up to 4096 tokens. It incorporates RMSNorm pre-normalization[Zhang and Sennrich, 2019], SwiGLU activation functions[Chowdhery *et al.*, 2022], rotary positional embeddings[Shaw *et al.*, 2018], and a grouped-query attention mechanism[Ainslie *et al.*, 2023] to achieve this efficient processing. We fine-tuned Llama-2-7B using a parameter-efficient fine-tuning technique (PEFT) called Dynamic Adaptation with Activation Memory Reduction (DyQLoRA-FA). The fine-tuning leveraged a vision-language instruction tuning dataset generated by GPT-4 Turbo with Vision, based on image captioning and open-ended VQA tasks. This task-specific fine-tuning enhances Llama-2's ability to comprehend complex language in niche domains, particularly evident in its improved interpretation of natural language questions related to electron micrographs. The resulting pretrained language model demonstrates advanced capabilities in question analysis and handling complex language, leading to a stronger correspondence between images and text. Llama-2's seamless integration with vision encoders makes it powerful for multimodal tasks. The proposed framework can effortlessly process both visual and textual data, which is particularly valuable when analyzing images and their corresponding descriptions.

## 1.3 Pretrained Large Multimodal Models

We build upon pre-trained Large Multimodal Models (LMMs) to generate image-question-answer triplets as instruction-tuning datasets to train smaller multimodal models (SMMs) through vision-language instruction tuning. This knowledge transfer, or distillation, from LMMs accelerates and enhances SMMs' learning, ultimately leading to more accurate, relevant, and contextually-aware responses in tasks demanding comprehension of both visual and linguistic inputs, such as zero-shot VQA and image captioning for electron microscopy images analysis. We utilize OpenAI's state-of-the-art multimodal model, GPT-4 Turbo with Vision (GPT-4-vision-preview), which surpasses the limitations of its predecessors, to efficiently generate high-quality training data for instruction tuning SMMs. This allows SMMs to generalize well to new, unseen questions. GPT-4 Turbo boasts a significantly expanded context window of 128k tokens ($\approx$ 300 pages per prompt), a 3x reduction in input token cost, a 2x reduction in output token cost, and a maximum output length of 4096 tokens for more elaborate text generation. The GPT-4 Turbo with Vision API, accessible through Multimodal Modeling as a Service (MMaaS), accepts both image and text inputs to generate multimodal outputs. By leveraging MMaaS, which utilizes proprietary GPT-4 Turbo with Vision as an on-demand cloud service accessed via an API, users can design task-specific prompts to query pre-trained LMMs for solving multimodal tasks of interest. This approach is analogous to how users ac-

cess LLMs via Language Modeling as a Service (LMaaS) for language-specific tasks. Designed for large-scale, concurrent requests, APIs are ideal for integration into automated systems. Our exploration of small multimodal models (SMMs) for electron micrograph analysis begins by leveraging GPT-4 Turbo with Vision (GPT-4V) to generate natural language questions as task-specific instructions for VQA and image-captioning tasks. By pairing these questions with the corresponding target electron micrographs, we create multimodal prompts that guide GPT-4V to generate contextually rich textual responses to natural language questions about the nanomaterial's structure and patterns underlying the electron micrographs. This approach capitalizes on GPT-4V's inherent domain-specific knowledge, acquired during training on a vast multimodal corpus, to yield comprehensive insights into these microscopic images. These insights helps to generate diverse multimodal instruction-following data, vital for training SMMs to generalize well on electron microscopy image analysis tasks.

## 1.4 Multimodal Instruction-Following Data

The generation of high-quality, diverse, and task-specific multimodal instruction-following data using GPT-4 Turbo with Vision is a powerful approach for training versatile, more efficient, and smaller multimodal models for VQA and image-captioning on microscopic image analysis tasks. This approach offers several benefits, including: (a) **Enhancing model capabilities**: GPT-4 Turbo with Vision's, owing to its vast pre-training knowledge can generate questions that comprehensively investigate diverse facets of nanomaterials underlying electron micrographs, including size, distribution, morphology, and structure. These questions are more complex, nuanced, and require reasoning and knowledge beyond basic image recognition. This can expand the limits of what smaller multimodal models can learn and enable them to answer more challenging visual questions about these microscopic images. (b) **Improving zero-shot learning**: Training smaller models on diverse questions and answers fosters deeper insights into the relationships between visual features, language, and task objectives. This enhances their ability to answer new questions on unseen microscopic images without further training, a critical element for practical applications. (c) **Facilitating knowledge distillation**: GPT-4 Turbo with Vision can generate detailed, nuanced question-answer pairs that describe microscopic images, including their visual properties such as shape, texture, patterns, and surface characteristics. Furthermore, it can draw connections to size, distribution, morphology, and structural relationships, leveraging its extensive internal knowledge acquired during pre-training. This facilitates knowledge distillation, transferring valuable task-specific knowledge from larger to smaller models. As a result, smaller models become more efficient, accurate, and transparent in their reasoning since they don't need to learn everything from scratch with expensive human-annotated datasets. (d) **Generating diverse question-answer pairs**: Finally, the end-user queries can be used to generate diverse question-answer pairs that delve into various aspects, properties, and characteristics of microscopic images. This further enriches the training data for smaller models, equipping them to handle a wider range of end-user queries. Our approach leverages the power of zero-shot chain-of-thought (CoT) prompting to guide large multimodal models (LMMs) like GPT-4 Turbo with Vision to create a novel training dataset of image-question-and-answer triples specifically designed for SMMs. As shown in Tables 11 - 20, the generated Q&A pairs correspond to a sample of microscopic images of different nanomaterials from the SEM dataset [Aversa *et al.*, 2018]. Through knowledge distillation, SMMs achieve performance on par with or even exceeding that of larger, more generalized multimodal models. The high-quality dataset, encompassing both images and corresponding Q&A pairs extracted from LMMs, provides a clear foundation for SMMs to understand how certain concept-based questions and their corresponding answers are visually represented.

**Prompt 1:** \*\*Basics\*\* - What type of nanomaterial is depicted in the image? - What is the scale of the image? (e.g., what does one unit of measurement represent?). **Prompt 2:** \*\*Morphology and Structure\*\* - What is the general shape or morphology of the nanomaterials in the image? - Are there distinct layers, phases, or domains visible? - Do the nanomaterials appear uniform in size and shape or are they varied?. **Prompt 3:** \*\*Size and Distribution\*\* - What is the approximate size or size range of the individual nanostructures? - How are the nanomaterials distributed throughout the image? (e.g., evenly spaced, clustered, random) - Is there any evidence of aggregation or bundling?. **Prompt 4:** \*\*Surface Characteristics\*\* - Does the nanomaterial appear smooth, rough, or have any specific textures? - Are there any visible defects, pores, or impurities on the surface?. **Prompt 5:** \*\*Composition and Elements\*\* - Is there evidence of compositional variations in the image (e.g., different colors, brightness, or contrasts)? - Are there any labels or markers indicating specific elements or compounds present?. **Prompt 6:** \*\*Interactions and Boundaries\*\* - How do individual nanostructures interact with one another? (e.g., are they touching, fused, or separate?) - Are there clear boundaries between different structures or phases?. **Prompt 7:** \*\*External Environment\*\* - Is there any evidence of the nanomaterial interacting with its surrounding environment or matrix (e.g., solvents, polymers, or other materials)? - Are there other structures or objects in the image that are not nanomaterials? If so, what are they?. **Prompt 8:** \*\*Image Technique and Modifications\*\* - What imaging technique was used to capture this image? (e.g., SEM, TEM) - Were there any post-processing or modifications made to the image (e.g., false coloring, 3D rendering)?. **Prompt 9:** \*\*Functional Features\*\* - If applicable, are there any functional features visible (e.g., active sites, regions with distinct properties)? - Are there dynamic processes captured in the image or is it a static representation?. **Prompt 10:** \*\*Context and Application\*\* - What is the intended application or use of the nanomaterial being depicted? - Is this a experimental sample, or a theoretical or simulation-based representation?

## 1.5 Vision Encoder

We start with an input image $\mathbf{I}$, a 3D tensor of dimensions $H \times W \times C$, representing height $H$, width $W$, and color channels $C$ per pixel. The image is divided into non-overlapping patches sized $P \times P \times C$. Tokenizing the image results in $n = \frac{HW}{P^2}$ patches. These patches are linearly encoded into 1D vectors, forming a sequence of tokens $\mathbf{I}' \in \mathbb{R}^{n \times d}$, where $d$ is the dimensionality of patch embeddings. Positional embeddings are added to each patch embedding to preserve spatial information. A special classification token, $<cls>$, is appended for aggregating information across patches for global representation. This token sequence is processed by a variant of the Vision Transformer (ViT) with stacked encoder layers using hierarchical attention mechanism. The stacked encoder layers process patch embeddings through higher-order attention mechanisms for multi-scale visual comprehension, from fine details to global context. It involves local and global multi-head attention phases, first focusing on patch interrelationships and then incorporating the classification token for a holistic understanding. The output is the embedding of the classification token $h_{cls}$, representing the image's unified visual context. In summary, the vision encoder breaks down the image into patches, converts them into tokens, and integrates them using a layered hierarchical attention mechanism to produce a comprehensive representation, $h_{cls}$, encapsulating both local and global aspects of the image. A vision encoder analyzes images to extract visual knowledge like objects, textures, and patterns, encoding them into a representation understood by a language model. This visual understanding is then fused with a natural language question, allowing the model to accurately interpret the question in the context of the image and generate precise answers to visual questions. This process effectively bridges the gap between visual and linguistic information, leading to richer and more meaningful multimodal reasoning and generation.

## 1.6 Sampling Strategies

To generate instruction-following multimodal data using GPT-4 Turbo with vision for few-shot image classification (refer to Figure 4) and to analyze electron micrographs for high intra-class dissimilarity, high inter-class similarity, and spatial heterogeneity (refer to Figures 5-7), we implement the following strategies. We train a Vision Transformer (ViT) through supervised learning to minimize cross-entropy loss and improve multiclass classification accuracy. The output embedding ($h_{cls}$) from the ($<cls>$) token provides a comprehensive image representation. For few-shot classification, we use a similarity-driven sampling method. We hypothesize that training with demonstrations that resemble the target image's data distribution will enhance adaptability and accuracy. To achieve this, we use cosine similarity of classification token embeddings to select the top-K similar images from the training set that are most similar to the target image. To comprehend high inter-class similarity and conversely, high intra-class dissimilarity, we generate question-answer pairs using GPT-4 Turbo with vision for each target image. For inter-class similarity, we sample highly similar images across nanomaterial categories. Conversely, for intra-class dissimilarity, we sample highly dissimilar images within the same

category. This process allows us to gain deeper insights from the electron micrographs.

## 1.7 Additional Information

We investigate the effect of using training data with diverse instruction lengths (image-question-answer triplets) generated by GPT-4 Turbo with Vision on the performance of smaller multimodal models. By incorporating both short (concise) and long (detailed) answers for the same natural language question into the training datasets, we aim to optimize these smaller models for tasks ranging from basic classification and image captioning to complex scenario analysis. This approach of employing varied-length data offers several potential benefits. Exposing a smaller model to diverse sentence structures and visual complexities fosters greater flexibility and adaptability. This approach enhances its ability to process real-world scenarios with varying levels of detail, improving generalizability and reducing overfitting. Furthermore, it challenges the smaller model's reasoning abilities, promoting a deeper understanding of the relationships between visual features and textual descriptions. Consequently, the smaller multimodal model's performance in tasks like image captioning and Visual Question Answering (VQA) improves, making it more robust and versatile for practical applications. Figures 3, 4, 5, 6, and 7 illustrate MAEMI, a multimodal assistant for electron micrograph analysis. MAEMI takes a multimodal prompt consisting of electron micrographs and supplementary information (e.g., metadata, annotations) and produces free-form text as output. Figure 3 and 4 show variants of the MAEMI framework on the zero/few-shot classification task. Figures 5, 6, and 7 illustrate how the MAEMI model can be adapted to address specific challenges in VQA tasks on electron micrographs, including intra-class dissimilarity, inter-class similarity, and spatial heterogeneity.

## 1.8 Experimental Setup

MAEMI is an AI assistant with an SMM (smaller multimodal model) as its backbone, specializing in electron microscopic image analysis. It integrates visual and textual data to understand microscopic images and answer questions. The SMM, with its vision and language capabilities, enables image captioning and visual question answering on microscopic images. The proposed vision-and-language assistant neural network architecture includes a vision encoder, a pretrained language-only-instruction-tuned decoder (Llama-2-7b), and multiple intertwined blocks of gated cross-attention and self-attention layers, allowing for task-specific adaptation on consumer hardware. This is achieved using the generated vision-language instruction-tuning data (image-text pairs) created by a large multimodal model (GPT-4 Turbo with Vision) to train the SMM for microscopic image analysis tasks. The smaller model leverages two key attention mechanisms: gated cross-attention and self-attention, to process both visual and textual data and generate human-like descriptions. Gated cross-attention allows the smaller model to selectively focus on relevant parts of the electron micrograph based on the textual input. Self-attention then refines the understanding by weighing different parts of the combined information. Despite its size, the smaller model generates accurate, contextually relevant,

and coherent text comparable to larger models, showcasing its ability to interpret natural language questions, utilize visual context, and produce effective responses. To train the SMM in a supervised learning setting, we employed the SEM dataset [Aversa *et al.*, 2018], a collection of electron micrographs of various nanomaterials with dimensions of $1024 \times 768 \times 3$ pixels. We preprocessed the microscopic images by resizing them to $224 \times 224 \times 3$ pixels and applying data standardization to normalize the data to have a mean of 0.5 and a variance of 1 across all channels. This preprocessing ensured that image values fell within the range of -1 and 1. To capture local features effectively, we divided the resized images into smaller patches, representing the micrographs as sequences of patches. Each patch was 32 pixels wide and high. We set both the patch dimension ($d_{\text{pos}}$) and the position embedding dimension ($d$) to 64 to capture sufficient spatial information within each patch sequence. This approach allowed the SMMs to learn from local features within the micrographs while maintaining context through the sequence of patches, improving the SMM's understanding and analysis of complex nanomaterials. Parameter-efficient fine-tuning of the Llama-2-7b model leverages the dynamic adaptation with activation memory reduction (DyQLoRA-FA) technique, characterized by three key hyperparameters: a) Rank ($r$): This parameter balances the smaller model's capacity and complexity by controlling the low-rank approximation of the trainable weight matrices. During training, $r$ is randomly selected from a predefined range ($r_{min} = 4, r_{max} = 16$). A higher rank yields a more expressive model with more adaptable parameters, while a lower rank promotes computational efficiency. (b) Alpha ($\alpha$): This scaling factor is typically set to a small value, such as $\frac{1}{r}$, based on the rank. Alpha controls the step size of the parameter updates. A larger alpha enables more aggressive updates, which can improve performance but may also cause training instability. (c) LoRA dropout: Specifically applied to low-rank adapter layers, this dropout mechanism combats overfitting and enhances generalization. A typical value for this hyperparameter is 0.05. We utilize 8-bit weight quantization for pre-trained model weights via the DyQLoRA-FA technique to enable efficient fine-tuning on consumer hardware while retaining comparable performance. The training regime for the SMM comprised 50 epochs, employing an initial learning rate of $1 \times 10^{-3}$ to ensure controlled optimization, and a batch size of 32. For the self-attention and cross-attention layers, we configured the number of attention heads (H) to be 4 and the dimensionality of Key/Query/Value ($d_h$) to be 32. To optimize SMM performance, we implemented two key strategies: (a) Early stopping on the validation set: We halted the training when the SMM's performance on the validation data plateaued, effectively preventing overfitting; (b) Learning rate scheduler: The learning rate was systematically reduced by half if the validation loss did not improve for five consecutive epochs. This reduction assisted the SMM in converging to a better solution and further mitigated overfitting. Furthermore, we employed the Adam optimization algorithm [Kingma and Ba, 2014] to update the SMM's trainable parameters. In our work, we have two types of instruction-following data: (a) a multi-class classification task - identification of nanomaterial category in zero/few shot settings, and (b) an open-ended visual question

answering (VQA) task. For supervised fine-tuning, we minimize the standard cross-entropy loss built using the PyTorch framework. We utilize Nvidia V100 GPUs (32GB RAM) to develop the custom SMM model.

## 1.9 Evaluation Metrics

In the field of image-captioning, visual question answering (VQA), several metrics are used to evaluate the quality of the generated text. These metrics assess different aspects of text generation, such as its similarity to reference texts, grammatical correctness, and semantic relevance. Here's an overview of some key metrics:

- **BLEU Score (Bilingual Evaluation Understudy)**: The BLEU score score evaluates machine-generated text quality by measuring its similarity to ground-truth references. It compares the overlapping n-grams (word sequences) between the translated text and reference texts, considering various n-gram lengths. BLEU mainly evaluates translation precision, ensuring the machine translation's words and phrases appear in the reference texts. It counts matching n-grams, using a clipping mechanism to avoid over-counting in cases of n-gram repetition. The score ranges from 0 to 1, with 0 indicating no overlap and 1 denoting complete similarity. Higher scores suggest better translation quality.

- **METEOR (Metric for Evaluation of Translation with Explicit Ordering)**: METEOR evaluates machine-generated text against ground-truth references, measuring overlap and considering linguistic qualities like synonymy and paraphrasing. It uses an alignment module to map unigrams between the candidate and reference texts, prioritizing exact matches, stem/lemma matching, and semantic similarity. To evaluate performance, it analyzes both how much of the reference text is addressed (coverage for recall) and how closely the generated text matches the wording (alignment for precision). Scores range from 0 to 1, with higher values indicating better performance. Unlike BLEU, METEOR better aligns with human quality judgments by considering recall, linguistic variations, and stronger correlation at the sentence or segment level.

- **ROUGE Score (Recall-Oriented Understudy for Gisting Evaluation)**: ROUGE measures the quality of generated text by comparing it with ground-truth references. It analyzes overlapping textual elements (like words or phrases) between the candidate and reference texts. The basic ROUGE-N metric computes the number of overlapping n-grams. Variants like ROUGE-L, ROUGE-W, and ROUGE-S measure the longest common subsequence, full word consecutive matches, and skip-bigram matches, respectively. Scores range from 0 to 1, where 0 means no overlap and 1 indicates complete overlap. Higher scores suggest better quality, showing the model's summary captures content similar to human references.

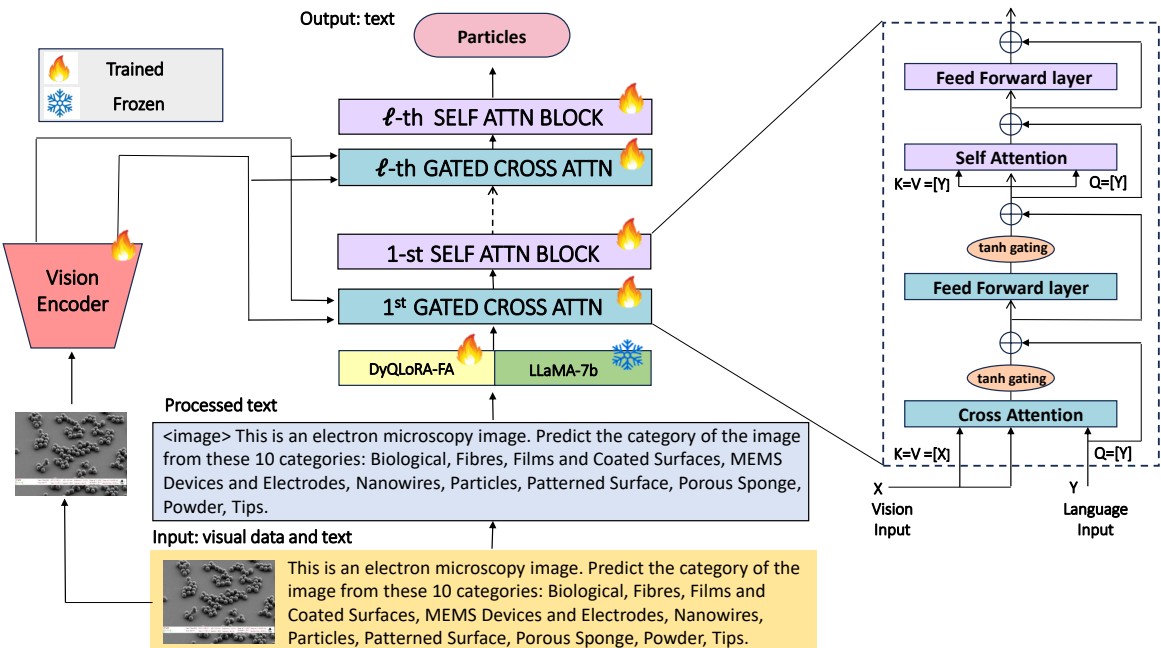

Figure 3: The schematic illustrates the small-scale, multimodal assistant for electron micrograph analysis (MAEMI), a content-aware, visually-conditioned, autoregressive text generation model that takes a multimodal prompt containing electron micrographs interleaved with textual descriptions, and produces free-form text as output. The input consists of a target image, user-provided supplementary text, and task-specific instruction. The goal is to categorize the image into one of ten categories in a zero-shot setting.

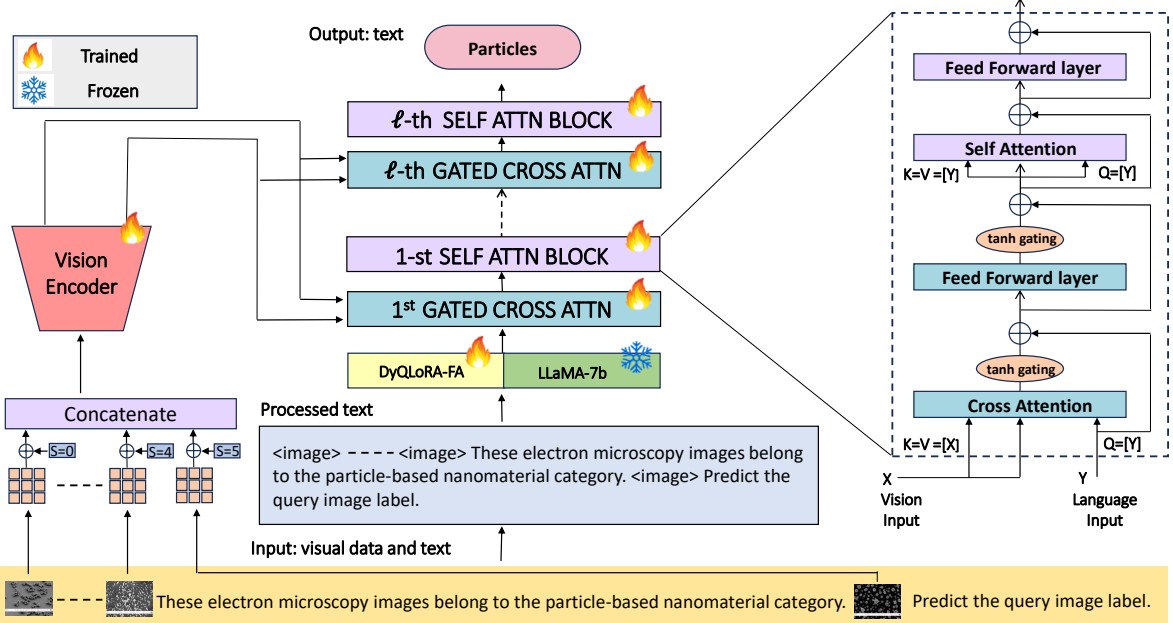

Figure 4: The schematic illustrates a small-scale, multimodal assistant for electron micrograph analysis (MAEMI), a visually-conditioned, autoregressive text generation model. The multimodal input consists of microscopic images arbitrarily interleaved with textual descriptions and produces free-form text as output. The input includes a few demonstration examples as input-output mappings(microscopic images their corresponding labels), and a task-specific instruction. The goal is to predict the label for the target image in a few-shot setting.

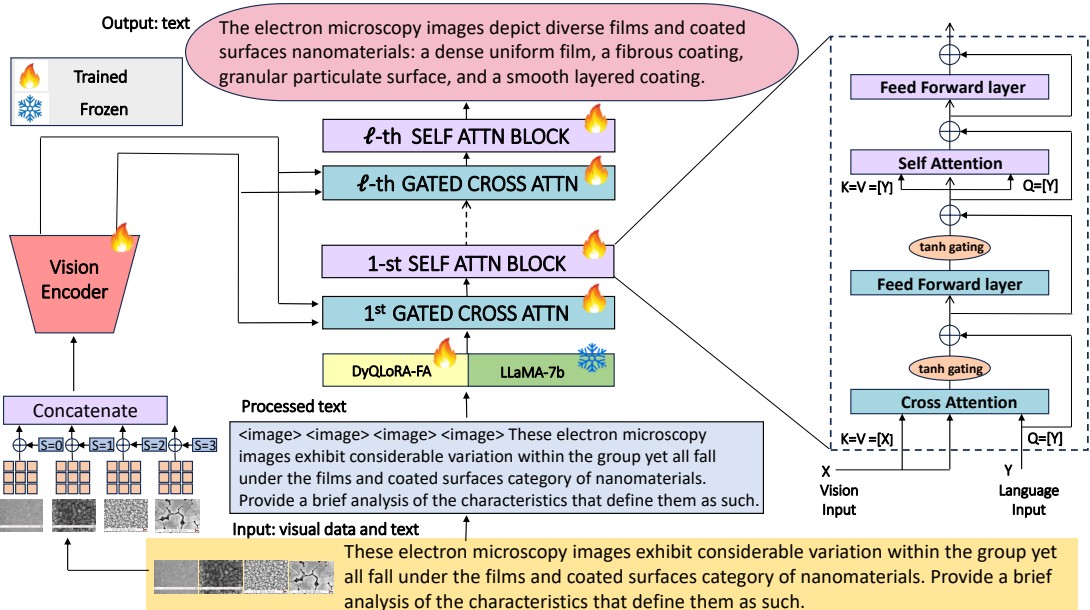

Figure 5: The schematic illustrates the proposed small-scale multimodal assistant for electron micrograph analysis (MAEMI). It leverages a multimodal prompt that interleaves visual data from electron microscopy images with user-provided auxiliary text data to generate descriptive output. The multimodal model is designed to generate accurate and concise descriptions of the visual features in high-contrast images, linking them to the classification of the electron micrographs into a specific nanomaterial category. During inference, MAEMI utilizes its domain-specific knowledge to interpret intertwined visual features and query text, generating accurate and informative responses about microscopic images within the specified category. Note: For clarity and brevity, the output text has been simplified.

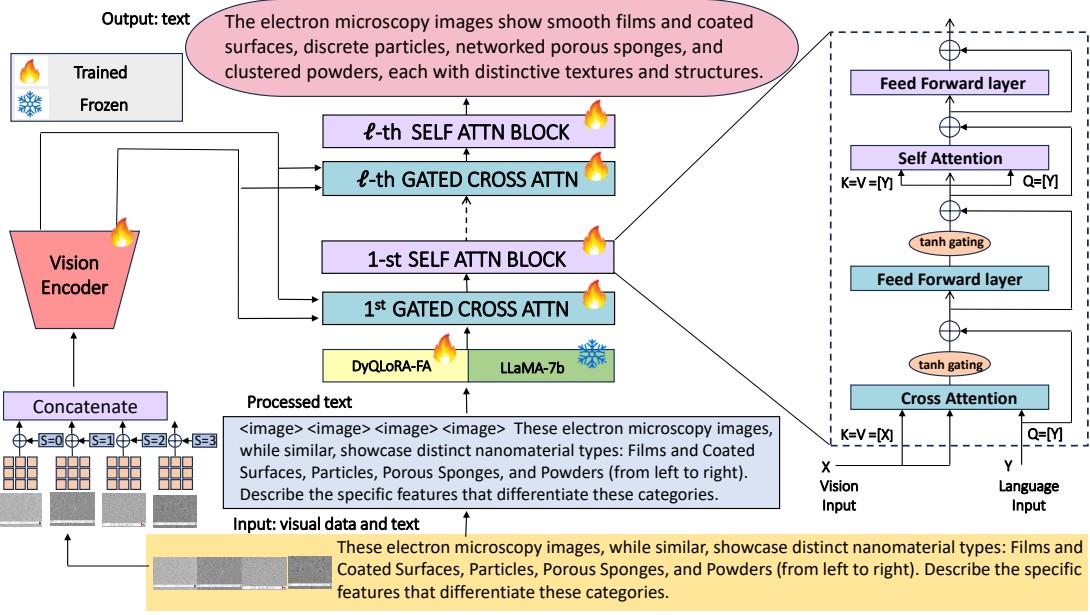

Figure 6: The schematic illustrates MAEMI, a proposed small-scale multimodal assistant for the VQA task on electron micrographs. It leverages a multimodal prompt that interleaves visual data of similar-looking, high-resolution electron micrographs showcasing diverse nanomaterial categories such as films and coated surfaces, particles, porous sponges, and powders with user-provided auxiliary text data. Additionally, MAEMI receives specific user queries that prompt it to analyze and describe the unique visual features distinguishing each category, thereby generating precise and concise responses describing the unique visual features distinguishing each category. Note: The output text is simplified for the sake of illustration and conciseness.

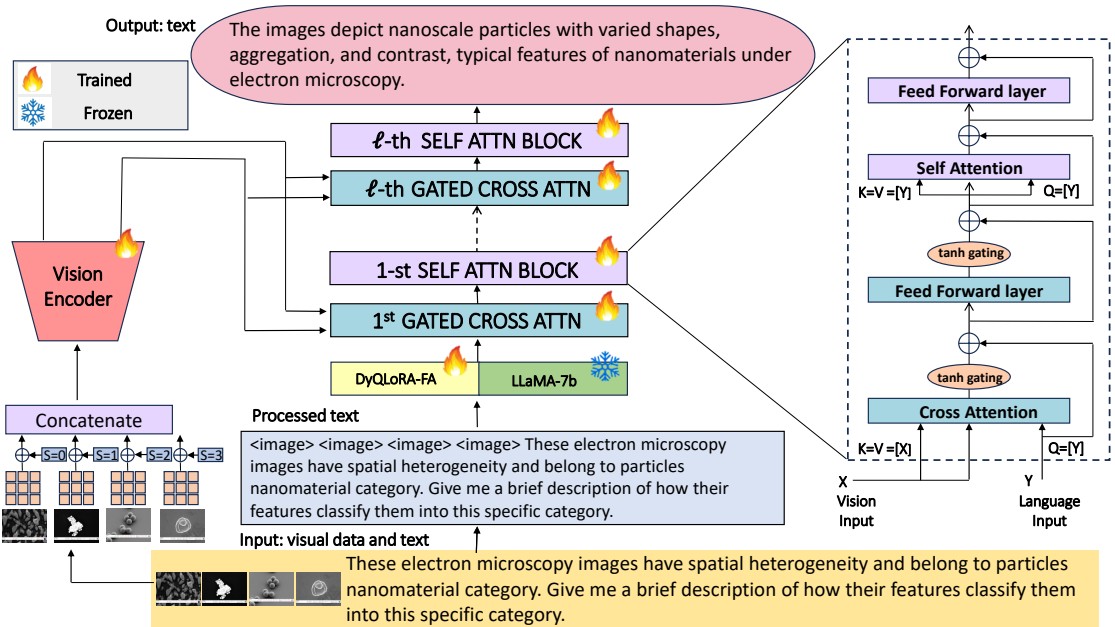

Figure 7: The schematic outlines the architecture of the small-scale multimodal assistant (MAEMI), which is tailored for the analysis of electron microscopy images of nanomaterials. It takes both visual and textual inputs: a series of high-resolution electron micrographs showcasing the spatial variations and diverse morphologies of the particles, combined with user-provided auxiliary text data. The multimodal model, guided by user instructions, produces brief, precise descriptions, highlighting the visual features unique to each nanomaterial category underlying the images. For clarity and brevity, the output text has been simplified. Note: We've presented the output text in a simplified format for better readability.

## 2 Experiments And Results

### 2.1 Datasets

Our study utilized the SEM dataset [Aversa et al., 2018], which comprises more than 21,000 electron micrographs covering ten different nanomaterials. We employed this comprehensive dataset to generate a diverse set of high-quality instruction-tuning data in the form of question-answer pairs using GPT-4 Turbo with Vision,. Figure 8 displays representative images for each of the ten nanomaterial categories. We trained our framework exclusively on this machine-generated multimodal data, eliminating the need for human-annotated data. In contrast to a previous study [Modarres et al., 2017], which worked with a subset of the data, we leveraged the entire publicly available dataset as the subset data was not publicly accessible in its entirety, enabling more comprehensive and robust framework training. We conducted rigorous benchmarking resulting in demonstrably improved task performance. Further experiments confirmed the framework's generalizability across open-source material datasets within its thematic area. Please refer to the technical appendix for more discussion.

## 3 Experiments

We evaluated our framework on zero-shot/few-shot multi-class classification tasks for microscopic images, image-captioning tasks, and open-ended VQA tasks. This in-depth analysis aimed to understand microscopic images better. Additionally, we conducted VQA tasks to assess intra-class dissimilarity, inter-class similarity, and spatial heterogeneity, providing a more comprehensive understanding of the nanomaterials underlying electron micrographs. In summary, we analyzed microscopic images using the proposed framework for classification, captioning, and answering questions. This improved understanding of image content and material properties.

### 3.1 Results

Our image captioning approach uses metrics like BLEU, METEOR, and ROUGE to evaluate text quality, focusing on aspects like similarity, language fluency, and coherence. As shown in Table 2, our framework, MAEMI, generates detailed and logically consistent captions, outperforming recent baselines like InstructBLIP[Dai et al., ], LLaVA[Liu et al., 2023], and MiniGPT-4[Zhu et al., 2023] on the image captioning task. Table 1 showcases representative electron microscope images with their true labels, alongside captions generated by our framework with their BLEU-2, ROUGE-L, and METEOR scores indicating caption similarity to the labels. Tables 8 and 9 present experimental results comparing the accuracy of our proposed multiclass classification framework against multiple baseline algorithms. Table 3 shows the framework's performance on open-ended VQA. Table 7 shows a sample of electron microscope images with true labels, generated captions, and similarity scores (BLEU-2, ROUGE-L, METEOR) comparing the captions to the labels. Sample questions and answers from the instruction-tuning Q&A dataset (created by GPT-4 Turbo with Vision) for training MAEMI are shown in Tables 11 - 20. Figures 5, 6, and 7 showcase tailored MAEMI variants for VQA tasks on electron micrographs, addressing intra-class dissimilarity, inter-class similarity, and spatial heterogeneity respectively. Tables 4, 5, and 6 compare the performance of different methods on the aforementioned VQA task, respectively.

Table 1: The table shows electron microscope images and their true captions alongside machine-generated captions. The table also includes evaluation metrics like BLEU-2, ROGUE-L, and METEOR, which measure the similarity between true captions and generated captions. By presenting both ground-truth and machine-generated captions side-by-side, the table enables analysis of the framework's performance in capturing visual details and semantics of the microscopic images. The multi-metric approach allows precise measurement of the proposed framework's performance on the captioning task for this scientific image dataset.

| Image | Ground Truth | Answers | BLEU-2/ ROGUE-L/ METEOR |
|---|---|---|---|
|  | This electron microscopy image displays a neuron with its dendritic tree and synaptic connections, magnified 10,000 times. | This electron microscopy image exhibits a neuron with its dendritic tree and synaptic connections, magnified 10,000 times | 0.847 0.944 0.941 |
|  | This SEM image shows tightly woven fibrous material, with each fiber distinctly magnified 225 times to reveal its twisted structure. | This SEM image displays tightly woven fibrous material, with every fiber distinctly magnified 225 times, revealing its twisted structure. | 0.659 0.821 0.852 |
|  | This SEM image captures a granular film surface with a magnification of 50,000 times, revealing the microstructure of the coated material. | This SEM image captures a granular film surface, magnified 50,000 times, revealing the microstructure of the coated material. | 0.724 0.878 0.767 |
|  | This SEM image shows a microelectromechanical system (MEMS) with intricate wiring and electrodes, captured at 100 times magnification. | This SEM image shows a microelectromechanical system (MEMS) with intricate wiring and electrodes, magnified 100 times | 0.795 0.882 0.842 |
|  | This SEM image depicts an array of vertical nanowires, showcasing their uniformity and high aspect ratio, magnified at 80,000 times. | This SEM image depicts an array of vertical nanowires, displaying their uniformity and high aspect ratio, magnified 80,000 times. | 0.843 0.927 0.902 |
|  | This SEM image reveals clusters of spherical nanoparticles, each grouping distinct from the others, magnified 11,000 times. | This SEM image shows clusters of spherical nanoparticles, each cluster distinct from the others, magnified 11,000 times | 0.813 0.889 0.879 |
|  | This SEM image displays a highly ordered, diamond-shaped patterned surface, magnified 345 times, characteristic of nano-fabrication techniques. | This SEM image displays a highly ordered, diamond-shaped patterned surface, magnified 345 times, typical of nano-fabrication techniques | 0.907 0.947 0.940 |
|  | This SEM image shows a porous sponge-like material with variously sized and shaped voids, magnified 50,000 times to reveal the texture and porosity. | This SEM image shows a porous sponge-like material with voids of various sizes and shapes, magnified 50,000 times, revealing the texture and porosity. | 0.616 0.760 0.778 |
|  | This SEM image reveals a dense aggregation of nanoscale particles forming a powder, with a magnification of 13,570 times. | This SEM image displays a dense aggregation of nanoscale particles forming a powder, magnified 13,570 times | 0.664 0.760 0.679 |
|  | This SEM image shows a sharply pointed nanomaterial tip, highlighted against a stark background at a magnification of 100,000 times. | This SEM image shows a sharply pointed nanomaterial tip, highlighted against a stark background, magnified 100,000 times. | 0.710 0.760 0.737 |

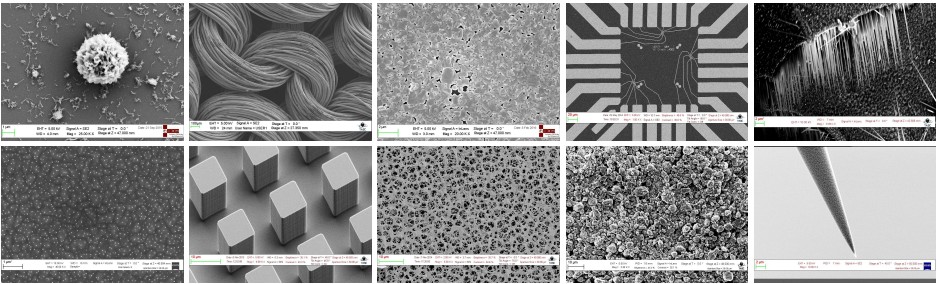

Figure 8: The figure shows representative microscopic images of diverse nanomaterials: biological structures, fibers, films, MEMS devices, nanowires (top); nanoparticles, patterned surfaces, porous sponges, powders, tips (bottom).

Table 2: The table summarizes the proposed framework's performance in comparison to various methods on the image captioning task.

| Method | BLEU-2 | BLEU-4 | ROUGE-1 | ROUGE-2 | ROUGE-L | METEOR |
|---|---|---|---|---|---|---|
| InstructBLIP[Dai *et al.,* ] | 0.7003 ± 0.032 | 0.6501 ± 0.039 | 0.8116 ± 0.016 | 0.7348 ± 0.005 | 0.8018 ± 0.021 | 0.8323 ± 0.024 |
| LLaVA[Liu *et al.,* 2023] | 0.7043 ± 0.035 | 0.6609 ± 0.043 | 0.8097 ± 0.016 | 0.7456 ± 0.005 | 0.8038 ± 0.021 | 0.8244 ± 0.023 |
| MiniGPT-4[Zhu *et al.,* 2023] | 0.7644 ± 0.086 | 0.6757 ± 0.100 | 0.8264 ± 0.035 | 0.7831 ± 0.014 | 0.8146 ± 0.047 | 0.8510 ± 0.052 |
| **MAEMI** | **0.7862 ± 0.089** | **0.6979 ± 0.115** | **0.9014 ± 0.041** | **0.8410 ± 0.016** | **0.8448 ± 0.054** | **0.8698 ± 0.062** |

Table 3: Table shows the performance of sLAVA compared to baselines on open-ended VQA task.

| Method | BLEU-2 (↑) | BLEU-4 (↑) | ROUGE-1 (↑) | ROUGE-2 (↑) | ROUGE-L (↑) | METEOR (↑) |
|---|---|---|---|---|---|---|
| InstructBLIP[Dai *et al.,* ] | 0.704±0.063 | 0.571±0.078 | 0.808±0.032 | 0.710±0.011 | 0.765±0.042 | 0.822±0.048 |
| LLaVA[Liu *et al.,* 2023] | 0.711±0.070 | 0.579±0.085 | 0.809±0.032 | 0.713±0.011 | 0.767±0.042 | 0.823±0.046 |
| MiniGPT-4[Zhu *et al.,* 2023] | 0.735±0.075 | 0.598±0.090 | 0.823±0.033 | 0.726±0.012 | 0.780±0.043 | 0.842±0.047 |
| **MAEMI** | **0.801±0.085** | **0.731±0.105** | **0.903±0.036** | **0.785±0.014** | **0.834±0.050** | **0.882±0.055** |

Table 4: The table shows sLAVA excels on VQA task on high intra-dissimilarity of nanomaterials.

| Method | BLEU-2 (↑) | BLEU-4 (↑) | ROUGE-1 (↑) | ROUGE-2 (↑) | ROUGE-L (↑) | METEOR (↑) |
|---|---|---|---|---|---|---|
| InstructBLIP[Dai *et al.,* ] | 0.667±0.063 | 0.541±0.078 | 0.764±0.032 | 0.672±0.011 | 0.724±0.042 | 0.778±0.048 |
| LLaVA[Liu *et al.,* 2023] | 0.651±0.070 | 0.530±0.085 | 0.740±0.032 | 0.652±0.011 | 0.702±0.042 | 0.754±0.046 |
| MiniGPT-4[Zhu *et al.,* 2023] | 0.673±0.075 | 0.548±0.090 | 0.754±0.033 | 0.664±0.012 | 0.714±0.043 | 0.770±0.047 |
| **MAEMI** | **0.732±0.085** | **0.668±0.105** | **0.826±0.036** | **0.717±0.014** | **0.762±0.050** | **0.807±0.055** |

Table 5: The table shows sLAVA excels on VQA task on high inter-similarity of nanomaterials.

| Method | BLEU-2 (↑) | BLEU-4 (↑) | ROUGE-1 (↑) | ROUGE-2 (↑) | ROUGE-L (↑) | METEOR (↑) |
|---|---|---|---|---|---|---|
| InstructBLIP[Dai *et al.,* ] | 0.676±0.063 | 0.548±0.078 | 0.775±0.032 | 0.682±0.011 | 0.734±0.042 | 0.789±0.048 |
| LLaVA[Liu *et al.,* 2023] | 0.675±0.070 | 0.550±0.085 | 0.767±0.032 | 0.677±0.011 | 0.730±0.042 | 0.782±0.046 |
| MiniGPT-4[Zhu *et al.,* 2023] | 0.690±0.075 | 0.561±0.090 | 0.773±0.033 | 0.682±0.012 | 0.733±0.043 | 0.791±0.047 |
| **MAEMI** | **0.744±0.085** | **0.679±0.105** | **0.841±0.036** | **0.730±0.014** | **0.775±0.050** | **0.820±0.055** |

Table 6: The table shows sLAVA excels on VQA task related to nanomaterials' spatial heterogeneity.

| Method | BLEU-2 (↑) | BLEU-4 (↑) | ROUGE-1 (↑) | ROUGE-2 (↑) | ROUGE-L (↑) | METEOR (↑) |
|---|---|---|---|---|---|---|
| InstructBLIP[Dai *et al.,* ] | 0.614±0.055 | 0.496±0.068 | 0.703±0.028 | 0.619±0.010 | 0.667±0.037 | 0.716±0.042 |
| LLaVA[Liu *et al.,* 2023] | 0.620±0.061 | 0.503±0.074 | 0.704±0.028 | 0.622±0.010 | 0.669±0.037 | 0.717±0.040 |
| MiniGPT-4[Zhu *et al.,* 2023] | 0.640±0.066 | 0.521±0.079 | 0.717±0.029 | 0.632±0.010 | 0.681±0.037 | 0.734±0.041 |
| **MAEMI** | **0.698±0.074** | **0.637±0.092** | **0.787±0.031** | **0.684±0.012** | **0.728±0.044** | **0.769±0.048** |

Table 7: The table shows a selection of electron microscope images with their corresponding true labels for an open-ended VQA task that describes the overall shape and morphology of the nanomaterials underlying the electron micrographs. We also include the framework generated responses or descriptions for each image. Additionally, the BLEU-2, ROGUE-L, and METEOR metrics are included to evaluate their similarity to the accurate labels.

| Image | Ground Truth | Answers | BLEU-2/ ROGUE-L/ METEOR |
|---|---|---|---|
|  | The nanomaterials exhibit a branched, web-like structure with varying strand thickness, indicative of a porous, high-surface-area morphology. | The nanomaterials exhibit a branched, web-like structure with varying strand thicknesses, suggesting a porous, high-surface-area morphology | 0.786 0.872 0.947 |
|  | The nanomaterials have a twisted, rope-like morphology with multiple strands intertwined together. | The nanomaterials possess a twisted, rope-like morphology, featuring multiple strands intertwined | 0.461 0.872 0.827 |
|  | The nanomaterials are polygonal, resembling a mosaic of tightly packed, irregularly shaped flat plates. | The nanomaterials appear polygonal, resembling a mosaic of tightly packed, irregular shaped flat plates. | 0.770 0.872 0.850 |
|  | The image showcases a microfabricated pattern with a square central area and symmetrically arranged geometric line patterns extending outward on a porous background. | This image showcases a microfabricated pattern with a square central area and symmetrically arranged geometric lines extending outward on a porous background. | 0.844 0.872 0.917 |
|  | The nanomaterials are cylindrical rods standing vertically with uniform alignment and consistent spacing between them. | The nanomaterials are cylindrical rods, standing vertically with uniform alignment and consistent spacing among them. | 0.787 0.872 0.861 |
|  | The nanomaterials exhibit rod-like and ellipsoidal shapes with smooth surfaces and are well-dispersed across the substrate. | The nanomaterials exhibit rod-like and ellipsoidal shapes, featuring smooth surfaces and are well-dispersed over the substrate | 0.736 0.872 0.808 |
|  | The nanomaterials have a hexagonal shape with well-defined edges and are arranged in an ordered, honeycomb-like pattern. | The nanoscale materials have a hexagonal shape with well-defined edges and are positioned in an orderly, honeycomb-like arrangement | 0.618 0.872 0.749 |
|  | The nanomaterials display a porous, foam-like structure with irregularly shaped voids and a network of interconnected struts. | The nanomaterials display a porous, foam-like structure, having irregularly shaped voids and a network of interconnected struts | 0.847 0.872 0.881 |
|  | The nanomaterials appear as clustered, irregularly shaped particles with a rough surface texture. | The nanomaterials are displayed as clustered, irregularly shaped particles with a rough surface textures. | 0.738 0.872 0.837 |
|  | The nanomaterial is conical with a pointed tip and a smooth gradient in diameter from base to apex. | The nanomaterial is conical, featuring a pointed tip and a smooth gradient in diameter from its base to apex. | 0.780 0.872 0.881 |

## 3.2 Empirical Insights

Our research thoroughly evaluated the proposed framework `MAEMI` for classifying electron micrographs of diverse nanomaterials. These complex materials vary in composition, morphology, structure, and other properties, which is evident in their electron micrographs. The framework achieved high accuracy on the imbalanced SEM dataset[Aversa *et al.*, 2018] using metrics like precision, recall, and F1-score, demonstrating its effectiveness in categorizing nanomaterials with different patterns in a zero-/few-shot setting. Table 10 reports the experimental results. The multi-metric approach provided a detailed analysis, highlighting `MAEMI`'s efficiency in handling various categories, especially those with fewer labeled instances. Overall, our findings confirm MAEMI's robustness in classifying nanomaterials, contributing to advancements in materials characterization and research.

Table 8: Table shows the performance comparisons: Our method vs. ConvNets, vision transformers (ViTs), & vision self-supervised learning(VSL) algorithms for classification task.

| | Algorithms | Top-1 | Top-5 |
|---|---|---|---|
| **ConvNets** | AlexNet([Krizhevsky *et al.*, 2017]) | 0.528 | 0.827 |
| | DenseNet([Huang *et al.*, 2017]) | 0.569 | 0.929 |
| | ResNet([He *et al.*, 2016]) | 0.485 | 0.897 |
| | VGG([Simonyan and Zisserman, 2014]) | 0.538 | 0.808 |
| | GoogleNet([Szegedy *et al.*, 2015]) | 0.609 | 0.969 |
| | SqueezeNet([Iandola *et al.*, 2016]) | 0.404 | 0.698 |
| **VSL** | Barlowtwins[Zbontar *et al.*, 2021] | 0.148 | 0.410 |
| | SimCLR[Chen *et al.*, 2020b] | 0.130 | 0.379 |
| | byol[Grill *et al.*, 2020] | 0.143 | 0.453 |
| | moco[He *et al.*, 2020] | 0.169 | 0.472 |
| | simsiam[Chen and He, 2021] | 0.188 | 0.535 |
| **Vision Transformers (ViTs)** | CCT[Hassani *et al.*, 2021] | 0.570 | 0.981 |
| | CVT[Wu *et al.*, 2021] | 0.577 | 0.930 |
| | ConViT[d'Ascoli *et al.*, 2021] | 0.609 | 0.957 |
| | ConvVT[Wu *et al.*, 2021] | 0.319 | 0.921 |
| | CrossViT[Chen *et al.*, 2021b] | 0.442 | 0.915 |
| | SwinT[Liu *et al.*, 2021] | 0.707 | 0.993 |
| | VanillaViT[Dosovitskiy *et al.*, 2020] | 0.655 | 0.970 |
| | Visformer[Chen *et al.*, 2021c] | 0.398 | 0.856 |
| | ATS[Fayyaz *et al.*, 2021] | 0.540 | 0.973 |
| | CaiT[Touvron *et al.*, 2021b] | 0.657 | 0.989 |
| | DeepViT[Zhou *et al.*, 2021] | 0.546 | 0.988 |
| | Dino[Caron *et al.*, 2021] | 0.049 | 0.437 |
| | Distillation[Touvron *et al.*, 2021a] | 0.533 | 0.955 |
| | LeViT[Graham *et al.*, 2021] | 0.624 | 0.970 |
| | NesT[Zhang *et al.*, 2022] | 0.660 | 0.985 |
| | PatchMerger[Renggli *et al.*, 2022] | 0.578 | 0.975 |
| | PiT[Heo *et al.*, 2021] | 0.555 | 0.979 |
| | RegionViT[Chen *et al.*, 2021a] | 0.606 | 0.948 |
| | SMIM[Xie *et al.*, 2021] | 0.171 | 0.646 |
| | T2TViT[Yuan *et al.*, 2021] | 0.749 | 0.992 |
| | ViT-SD[Lee *et al.*, 2021] | 0.597 | 0.973 |
| | Zero-Shot-Image Captioning / `MAEMI` | **0.773** | **0.876** |
| | Few-Shot-Image Captioning / `MAEMI` | **0.965** | **0.991** |

Table 9: The table shows the comparison of supervised-learning GNNs(Graph Neural Networks), self-supervised GCL(Graph Contrasting Learning) algorithms on the classification task.

| | Algorithms | Top-1 | Top-5 |
|---|---|---|---|
| **GCL** | GBT[Bielak *et al.*, 2021] | 0.547 | 0.706 |
| | GRACE[Zhu *et al.*, 2020] | 0.598 | 0.750 |
| | BGRL[Thakoor *et al.*, 2021] | 0.556 | 0.696 |
| | InfoGraph[Sun *et al.*, 2019] | 0.526 | 0.702 |
| **Graph Neural Networks** | APPNP[Klicpera *et al.*, 2018] | 0.632 | 0.786 |
| | AGNN[Thekumparampil *et al.*, 2018] | 0.538 | 0.894 |
| | ARMA[Bianchi *et al.*, 2021] | 0.582 | 0.987 |
| | DNA[Fey, 2019] | 0.622 | 0.916 |
| | GAT[Veličković *et al.*, 2017] | 0.491 | 0.985 |
| | GGConv[Li *et al.*, 2015] | 0.563 | 0.992 |
| | GraphConv[Morris *et al.*, 2019] | 0.658 | 0.996 |
| | GCN2Conv[Chen *et al.*, 2020a] | 0.732 | 0.998 |
| | ChebConv[Defferrard *et al.*, 2016] | 0.504 | 0.951 |
| | GraphConv[Morris *et al.*, 2019] | 0.509 | 0.993 |
| | GraphUNet[Gao and Ji, 2019] | 0.657 | 0.978 |
| | MPNN[Gilmer *et al.*, 2017] | 0.603 | 0.999 |
| | RGGConv[Bresson and Laurent, 2017] | 0.618 | 0.961 |
| | SuperGAT[Kim and Oh, 2022] | 0.598 | 0.985 |
| | TAGConv[Du *et al.*, 2017] | 0.598 | 0.999 |
| | Zero-Shot-Image Captioning / `MAEMI` | **0.773** | **0.876** |
| | Few-Shot-Image Captioning / `MAEMI` | **0.965** | **0.991** |

## 4 Conclusion

Our research unveils a groundbreaking method for analyzing electron micrographs for the semiconductor industry. We utilize transfer learning to distill knowledge, customizing an instruction-following language-vision assistant trained on a unique multimodal data created with GPT-4 Turbo for VQA tasks on consumer hardware. The pre-trained assistant allows further customization with private data, all without exposing sensitive information to external, proprietary multimodal models. This secure, efficient, and cost-effective methodology unlocks exciting possibilities for enterprise applications. Empirical results confirm our framework's superiority, achieving notable accuracy improvements over prior techniques while

| Category | Multi-class metrics | | |
|---|---|---|---|
| | **Precision** | **Recall** | **F1 Score** |
| Biological | 0.949±0.009 | 0.981±0.007 | 0.954±0.013 |
| Tips | 0.939±0.005 | 0.952±0.008 | 0.936±0.011 |
| Fibres | 0.982±0.007 | 0.987±0.000 | 0.982±0.000 |
| Porous Sponge | 0.956±0.014 | 0.955±0.013 | 0.955±0.010 |
| Films Coated Surface | 0.961±0.005 | 0.960±0.009 | 0.961±0.008 |
| Patterned Surface | 0.969±0.016 | 0.968±0.006 | 0.953±0.014 |
| Nanowires | 0.953±0.012 | 0.965±0.007 | 0.977±0.011 |
| Particles | 0.961±0.006 | 0.959±0.011 | 0.942±0.023 |
| MEMS Devices | 0.965±0.011 | 0.953±0.008 | 0.953±0.009 |
| Powder | 0.959±0.014 | 0.960±0.009 | 0.940±0.011 |

Table 10: The table shows the effectiveness of our proposed framework, compared to existing methods, in terms of precision, recall, and F1-score for accurately classifying nanomaterials of different categories.

remaining computationally efficient.

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

Table 11: The table presents an instruction-following dataset composed of image-question-answer triples generated using GPT-4 Turbo with vision in response to open-ended natural language prompts about a microscopic image of patterned surface nanomaterials. In summary, this GPT-assisted vision-language dataset aids in the tuning of smaller models for the analysis and interpretation of complex microscopic images in the context of open-ended VQA tasks.

**Based on the provided image of patterned surface nanomaterials, please answer the following questions.**

| Query prompt | GPT-4 Turbo with vision output |
| --- | --- |
| **Basics** | The image is a scanning electron microscope (SEM) image of a patterned surface nanomaterial, with a scale bar indicating that 1 micrometer ($\mu$m) is the unit of measurement. |
| **Morphology and Structure** | The nanomaterials in the image have a hexagonal shape or morphology. There do not appear to be distinct layers, phases, or domains within the individual hexagonal units, but there is a consistent pattern to their arrangement. The nanomaterials appear to be uniform in size and shape, suggesting a controlled fabrication process. |
| **Size and Distribution** | The nanostructures are hexagonal and uniform in size, likely in the hundreds of nanometers range, judging by the 1 μm scale bar. They are evenly spaced throughout the image with no evidence of aggregation or bundling. |
| **Surface Characteristics** | The nanomaterial appears to have a textured surface, possibly due to the patterning technique used to create the hexagonal shapes. No visible defects, pores, or impurities are apparent in the image provided, suggesting a clean and uniform fabrication process. |
| **Composition and Elements** | The SEM image is monochromatic, with brightness and contrast variations hinting at topographical or compositional differences. The absence of specific labels or markers means exact composition can't be determined. Annotations are standard SEM parameters, not compositional data. |
| **Interactions and Boundaries** | The individual nanostructures are separate, with clear boundaries between them, and do not appear to be touching or fused. There is a consistent gap that defines the boundaries of each hexagonal structure, indicating well-defined individual units. |
| **External Environment** | The SEM image displays isolated nanomaterials, free from interactions with solvents, polymers, or other substances. The uniform, featureless background indicates either a vacuum or a non-descript solid substrate at this magnification, with no other visible structures or objects unrelated to the patterned nanomaterials. |
| **Image Technique and Modifications** | The image was taken with a Scanning Electron Microscope (SEM), evident from typical SEM annotations like "EHT" and "WD". It shows no signs of post-processing, such as false coloring, and appears as a standard monochromatic SEM output. |
| **Functional Features** | There are no obvious functional features like active sites with distinct properties visible in this image. |
| **Context and Application** | The intended application of the nanomaterial is unclear without context. The image is a real SEM capture of an experimental sample, not a theoretical or simulation-based representation. |

Table 12: The table presents an instruction-following dataset, comprising image-question-answer triples generated using GPT-4 Turbo with vision in response to open-ended natural language prompts about a microscopic image depicting biological nanomaterials. In summary, this GPT-assisted vision-language dataset facilitates the training of smaller models, enabling them to effectively analyze complex microscopic images for open-ended VQA tasks.

**Based on the provided image of biological nanomaterials, please answer the following questions.**

| Query prompt | GPT-4 Turbo with vision output |
|---|---|
| **Basics** | The biological nanomaterial shown in the image is a neuron with its dendritic tree. The scale bar represents two micrometers (2 $\mu$m), providing a reference for the size of the structures in the image. |
| **Morphology and Structure** | The nanomaterials in the image are biological in nature and exhibit a complex morphology characteristic of neuronal cells. The neuron has a prominent cell body from which multiple dendrites extend, forming an intricate network. There are no distinct layers, phases, or domains in the traditional material science sense, as this is a biological sample. The neuron displays compartmentalization with the cell body, dendrites, and possibly axon hillocks or initial segments visible. The nanomaterials are varied in size and shape, as typical for biological structures. Dendrites exhibit diverse thickness, length, and branching patterns, reflecting inherent biological variability. |
| **Size and Distribution** | The individual nanostructures, the dendrites, vary in size but typically range from a few hundred nanometers to a couple of micrometers in diameter. They are distributed throughout the image in a connected, somewhat random pattern, emanating from a central neuron cell body. There is no evidence of aggregation or bundling, as these structures tend to spread out rather than cluster together. |
| **Surface Characteristics** | The nanomaterial, which is a neuron with its dendritic tree, appears to have a textured surface characteristic of biological structures, rather than being smooth. The dendrites and cell body show a complex, granular surface texture under the magnification provided. No specific defects, pores, or impurities can be discerned from this image; such features would require higher magnification to resolve. |
| **Composition and Elements** | The monochromatic image, typical of electron microscopy, lacks color-based compositional variations. Yet, it shows variations in brightness and contrast, suggesting topographical or material density differences in the neuron and its dendrites. |
| **Interactions and Boundaries** | The monochromatic electron microscope image shows topographical and density variations, but no compositional differences or labeled elements. |
| **External Environment** | The SEM image shows only the isolated neuron, with no evidence of external interactions or matrix material. |
| **Image Technique and Modifications** | The image was captured using Scanning Electron Microscopy (SEM). There is no evident post-processing or modifications like false coloring or 3D rendering; it appears to be a standard grayscale SEM image |
| **Functional Features** | The SEM image of the neuron does not visibly show any specific functional features like active sites. |
| **Context and Application** | The image is of an experimental sample, an actual neuron, captured using Scanning Electron Microscopy (SEM), rather than a theoretical or simulation-based representation. |

1157

Table 13: The table presents an instruction-following dataset composed of image-question-answer triples generated using GPT-4 Turbo with vision in response to open-ended natural language prompts about a microscopic image depicting fibers nanomaterial. In summary, this GPT-assisted vision-language dataset facilitates the training of smaller models, enabling them to effectively analyze and interpret complex microscopic images for open-ended VQA tasks.

**Based on the provided image of fibres nanomaterials, please answer the following questions.**

| Query prompt | GPT-4 Turbo with vision output |
|---|---|
| **Basics** | The image shows nanofibers, which are extremely fine fibers with diameters in the nanometer range. The scale of the image is not indicated, so it's not possible to determine the exact measurement without a scale reference. However, given that these are nanofibers, we can infer that the width of these fibers is likely in the range of tens to hundreds of nanometers. |
| **Morphology and Structure** | The nanomaterials in the image have a fibrous morphology, indicating a long, slender shape. There are no distinct layers, phases, or domains visible in the image provided. The fibers seem fairly uniform in shape, suggesting a controlled synthesis process. However, due to the image's resolution and the absence of a scale, it is challenging to accurately comment on the uniformity of size. |
| **Size and Distribution** | The fibers appear to be in the range of nanometers in diameter, although the exact size cannot be determined without a scale. They are densely distributed, with some bundling and clustering visible. |
| **Surface Characteristics** | The nanomaterial fibers appear to have a smooth texture overall. No clear defects, pores, or impurities are visible from this image. The resolution is not sufficient to identify finer surface details. |
| **Composition and Elements** | The image is in grayscale, and there are no visible labels or markers indicating specific elements or compounds. There are no apparent color variations that would suggest compositional differences. The uniformity of contrast throughout the fibers suggests they are likely composed of a single material or that any compositional variations are not discernible at this scale and image quality. |
| **Interactions and Boundaries** | The nanostructures appear to be touching and interacting closely, with some degree of fusion or entanglement, typical for fibrous networks. There are no clear boundaries between different structures or phases; it looks like a continuous phase of similar fibers. |
| **External Environment** | The SEM image shows isolated nanofibers with no visible external interactions or matrix. |
| **Image Technique and Modifications** | The grayscale image, likely captured via a standard scanning electron microscope (SEM), showcases nanomaterials in sharp detail, typical of SEM outputs. No evidence suggests post-processing, like false coloring or 3D rendering, indicating a straightforward SEM image without modifications. |
| **Functional Features** | The image does not show any distinct functional features like active sites or regions with varying properties; it appears to be a uniform structure. Since it's an SEM image, it captures a static representation of the nanofibers and does not depict any dynamic processes. |
| **Context and Application** | The intended application of the nanomaterial in the image cannot be determined without additional context. The image appears to be of an experimental sample, typical of real-world SEM imaging, rather than a theoretical or simulation-based representation. |

1158

Table 14: The table presents an instruction-following dataset composed of image-question-answer triples generated using GPT-4 Turbo with vision in response to open-ended natural language prompts about a microscopic image of films and coated surfaces nanomaterials. In summary, this GPT-assisted vision-language dataset aids in the tuning of smaller models for the analysis and interpretation of complex microscopic images in the context of open-ended VQA tasks.

**Based on the provided image of films and coated surface nanomaterials, please answer the following questions.**

| Query prompt | GPT-4 Turbo with vision output |
|---|---|
| **Basics** | The image is a Scanning Electron Microscope (SEM) image of a nanomaterial, possibly metallic or ceramic nanoparticles. The scale bar indicates that 1 micrometer ($\mu m$) is represented, allowing for the size of the particles to be estimated in relation to this scale. |
| **Morphology and Structure** | The nanomaterials in the image have a general morphology that appears to be polygonal, with most particles exhibiting angular edges and flat surfaces. There are no distinct layers, phases, or domains visible; it appears to be a homogenous surface of similar particles. The particles seem to be fairly uniform in size and shape, suggesting a controlled synthesis process. |
| **Size and Distribution** | The individual nanostructures appear to be in the sub-micron size range, likely several hundred nanometers across, given the scale bar of 1 micrometer for reference. They are distributed throughout the image quite densely, with little to no space between adjacent particles, which indicates a uniform surface coverage. There seems to be minimal aggregation or bundling; the particles mostly appear as distinct entities with some contact points, which is typical for particles that have settled out of a dispersion onto a substrate. |
| **Surface Characteristics** | The nanomaterial surface appears to be relatively rough with a textured appearance, which is characteristic of aggregated nanoparticles. There are no obvious defects, pores, or impurities visible at this magnification. The roughness is likely due to the individual particles' edges and surfaces rather than a feature of the particles themselves. |
| **Composition and Elements** | There is no evidence of compositional variations in the grayscale SEM image, and there are no labels indicating specific elements or compounds. The uniformity in color suggests a homogenous material, but specific composition cannot be determined from this image alone. |
| **Interactions and Boundaries** | The individual nanostructures are touching and seem to be sintered, with no clear gaps between them. There are clear boundaries between individual particles, but no distinct phases or separate structures are visible. |
| **External Environment** | No, there is no evidence of the nanomaterial interacting with any surrounding environment or matrix in the SEM image, and no other structures or objects are present other than the nanomaterials themselves. |
| **Image Technique and Modifications** | The image was captured using a Scanning Electron Microscope (SEM). There are no indications of post-processing or modifications such as false coloring or 3D rendering; it appears to be a standard grayscale SEM image. |
| **Functional Features** | In the SEM image, functional features like active sites or regions with distinct properties are not discernible. SEM images typically provide a static representation and cannot capture dynamic processes, so this image is a static representation of the nanomaterial at the time of imaging. |
| **Context and Application** | From the image alone, the nanomaterial's purpose is indeterminate. It's a real, experimental Scanning Electron Microscope (SEM) image, used for analyzing surface morphology and structure. |

Table 15: The table presents an instruction-following dataset composed of image-question-answer triples generated using GPT-4 Turbo with vision in response to open-ended natural language prompts about a microscopic image of nanowires nanomaterials. In summary, this GPT-assisted vision-language dataset aids in the tuning of smaller models for the analysis and interpretation of complex microscopic images in the context of open-ended VQA tasks.

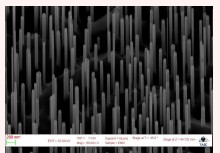

**Based on the provided image of nanowire nanomaterials, please answer the following questions.**

| Query prompt | GPT-4 Turbo with vision output |
| --- | --- |
| **Basics** | The image depicts nanowires, and the scale bar represents 200 nanometers (nm). |
| **Morphology and Structure** | The nanomaterials have a cylindrical, rod-like morphology, typical of nanowires. There are no distinct layers, phases, or domains visible in this image. The nanowires appear to be relatively uniform in size and shape, suggesting a controlled synthesis process. |
| **Size and Distribution** | The individual nanostructures, which are nanowires, are approximately 200 nanometers or less in length as indicated by the scale bar. They appear to be densely packed and evenly distributed across the image with no significant clustering or random distribution. There is no clear evidence of aggregation or bundling; the nanowires stand individually, which is typical for well-synthesized nanowires. |
| **Surface Characteristics** | The nanomaterials, which are nanowires, appear to have a smooth surface with no significant roughness or specific texture visible at this magnification. There are no clearly visible defects, pores, or impurities on the surface of the nanowires in the provided image. The uniformity and smoothness suggest a high level of control in the synthesis process of these nanostructures. |
| **Composition and Elements** | In the grayscale electron microscope image, variations in brightness or contrast usually show compositional differences, due to varied electron scattering by different elements. The provided image's uniform color and contrast imply a homogeneous composition in the nanowires. No labels or markers indicate specific elements or compounds, and the image doesn't directly reveal the nanomaterials' chemical composition. |
| **Interactions and Boundaries** | The individual nanostructures, which are nanowires, appear separate and not fused together, with each nanowire standing distinct from others. There are clear boundaries between each nanowire, with no visible merging or blending at the bases or tips, indicating they are well-defined individual entities. |
| **External Environment** | There is no visible evidence in the image of the nanomaterial interacting with an external environment or matrix such as solvents, polymers, or other materials. The image appears to be focused solely on the nanowires themselves. There are no other structures or objects present that are distinguishable from the nanomaterials. |
| **Image Technique and Modifications** | The image was captured using Scanning Electron Microscopy (SEM). There are no indications of post-processing or modifications like false coloring or 3D rendering; it appears to be a standard grayscale SEM image. |
| **Functional Features** | There are no visible functional features like active sites or regions with distinct properties in the image. The image is a static representation; it does not capture dynamic processes, as SEM images are snapshots and cannot depict motion or change over time. |
| **Context and Application** | Nanomaterial application unclear. Depicted nanowires used in electronics, photonics, research. SEM image suggests experimental sample, not theoretical representation, for observation at high magnifications. |

Table 16: The table presents more informative and accurate question-answer pairs, compared to traditional methods, generated using GPT-4 Turbo with vision in response to natural language prompts about an input microscopic image from the porous sponge nanomaterial category. To summarize, GPT-assisted visual instruction data generation (that is, question-answer pairs) provides in-depth insights into an image related to porous sponge nanomaterials.

**Based on the provided image of porous sponge nanomaterials, please answer the following questions.**

| Query prompt | GPT-4 Turbo with vision output |
|---|---|
| **Basics** | The nanomaterial in the image is a porous sponge-like structure. The scale of the image indicates that 1 micrometer ($\mu$m) is the unit of measurement. |
| **Morphology and Structure** | The nanomaterials in the image have an irregular, highly porous morphology with many voids. There are no distinct layers, phases, or domains visible; it's a continuous network of material. The nanomaterials do not appear uniform in size and shape; there's a variety in the pore sizes and the thickness of the material between the pores. |
| **Size and Distribution** | The individual nanostructures vary in size, with pores ranging from very small to relatively large in the context of the scale provided. The pores do not have a uniform size and seem to be randomly distributed throughout the material. There is no clear evidence of aggregation or bundling; the material seems to be a single continuous phase with interconnected pores. |
| **Surface Characteristics** | The nanomaterial appears to have a rough texture with numerous visible pores of varying sizes, which are characteristic of a highly porous surface. The walls of the material between the pores also seem rough and irregular. Visible defects or variations in the pore size and distribution across the surface can be considered impurities, which are typical for porous materials like aerogels or foam-like structures. |
| **Composition and Elements** | There are no visible compositional variations in the image; it appears uniform in color and contrast. No labels indicate specific elements or compounds within the nanomaterial. |
| **Interactions and Boundaries** | The individual nanostructures are fused, forming a continuous network without clear boundaries between separate structures or phases. The pores are interconnected, suggesting a single, cohesive material. |
| **External Environment** | The image does not show any evidence of the nanomaterial interacting with an external environment or matrix such as solvents or polymers. The entire field of view is occupied by the nanomaterial, with no other distinct structures or objects present that are not part of the nanomaterial itself. All visible features appear to be intrinsic to the porous structure of the nanomaterial. |
| **Image Technique and Modifications** | The imaging technique used is Scanning Electron Microscopy (SEM), as indicated by the labels on the image such as "EHT" (electron high tension), which is specific to SEM. There are no obvious signs of post-processing or modifications such as false coloring; the image appears to be a SEM grayscale representation. |
| **Functional Features** | As for functional features like active sites or regions with distinct properties, these are not discernible at the resolution and format of the image. |
| **Context and Application** | The nanomaterial is likely intended for applications like catalysis, filtration, or insulation, given its porous structure. The image is of an actual experimental sample, as indicated by its SEM imaging characteristics. |

Table 17: The table presents an instruction-following dataset composed of image-question-answer triples generated using GPT-4 Turbo with vision in response to open-ended natural language prompts about a microscopic image of tips nanomaterials. In summary, this GPT-assisted vision-language dataset aids in the tuning of smaller models for the analysis and interpretation of complex microscopic images in the context of open-ended VQA tasks.

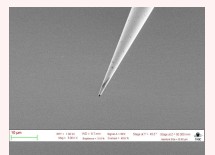

**Based on the provided image of tips nanomaterials, please answer the following questions.**

| Query prompt | GPT-4 Turbo with vision output |
| --- | --- |
| **Basics** | The image shows an electron microscope image of a nanomaterial tip, likely for scanning probe microscopy. The scale bar represents 10 micrometers. |
| **Morphology and Structure** | The nanomaterial in the image has a conical shape, characteristic of a sharp tip used in microscopic probes. There are no distinct layers, phases, or domains visible in this image. It appears to be a single, uniform structure without visible variation in size or shape along its length, suggesting a consistent manufacturing process. |
| **Size and Distribution** | The image depicts a single nanostructure with a conical tip. The scale bar indicates that the tip width is less than 10 micrometers, and the tip itself tapers to a point, which is characteristic of nanomaterial tips used for scanning probe microscopy. There is only one nanomaterial visible, so we cannot comment on the distribution or spacing of multiple nanostructures. Similarly, there is no evidence of aggregation or bundling, as the image shows a solitary nanostructure. |
| **Surface Characteristics** | The nanomaterial tip in the image appears relatively smooth, which is typical for nanomaterials used in high-precision applications like microscopy. There are minor variations on the surface that could be indicative of some roughness or surface texture, but these are minimal. There don't seem to be any large visible defects, pores, or impurities; however, the presence of small surface imperfections cannot be ruled out at this magnification and resolution. |
| **Composition and Elements** | The grayscale image does not show compositional variations in color. However, brightness and contrast variations can indicate differences in composition or density. The uniform grayscale suggests homogeneous material composition. There are no labels indicating specific elements or compounds. |
| **Interactions and Boundaries** | The image shows a single nanomaterial tip with no visible interactions or boundaries with other nanostructures. |
| **External Environment** | In the image, there is no direct evidence of the nanomaterial interacting with a surrounding environment or matrix. The background is featureless and uniform, which suggests that the image focuses solely on the nanomaterial tip and does not include other materials or structures such as solvents, polymers, or additional objects. |
| **Image Technique and Modifications** | The image was captured using a Scanning Electron Microscope (SEM). There are no apparent post-processing or modifications like false coloring or 3D rendering; it's a raw grayscale SEM image. |
| **Functional Features** | No specific functional features like active sites are visible in the image. It's a static representation, not capturing any dynamic processes. |
| **Context and Application** | The nanomaterial depicted is likely for use in scanning probe microscopy, such as Atomic Force or Scanning Tunneling Microscopy. The image is of an experimental sample. |

Table 18: The table presents an instruction-following dataset composed of image-question-answer triples generated using GPT-4 Turbo with vision in response to open-ended natural language prompts about a microscopic image of particles nanomaterials. In summary, this GPT-assisted vision-language dataset aids in the tuning of smaller models for the analysis and interpretation of complex microscopic images in the context of open-ended VQA tasks.

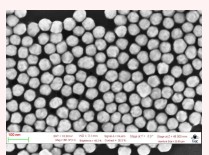

**Based on the provided image of particles nanomaterials, please answer the following questions.**

| Query prompt | GPT-4 Turbo with vision output |
| --- | --- |
| **Basics** | The image shows spherical nanoparticles, and the scale bar at the bottom indicates that 100 nanometers (nm) is the unit of measurement. Each particle appears to be about 100 nm in diameter. |
| **Morphology and Structure** | The nanomaterials in the image are generally spherical in shape and appear to be highly uniform in size and shape, suggesting a monodisperse distribution. There are no distinct layers, phases, or domains visible; the particles are consistent in their morphology throughout the image. |
| **Size and Distribution** | The approximate size of the individual nanostructures, judging by the scale bar, is about 100 nanometers in diameter. The nanomaterials are distributed evenly throughout the image, with a uniform spacing that suggests a well-ordered array and minimal aggregation or bundling. There's no visible evidence of clustering; each particle is isolated from its neighbors, indicative of a stable colloidal system. |
| **Surface Characteristics** | The nanomaterials appear to have a smooth surface without any noticeable roughness or specific textures. The resolution of the image does not allow for the identification of any defects, pores, or impurities on the surface of the nanoparticles. The overall uniformity and smoothness suggest high-quality synthesis methods without obvious defects or contamination. |
| **Composition and Elements** | There are no visible compositional variations in the image, and there are no labels indicating specific elements or compounds. The particles appear uniformly bright, suggesting a homogenous material without visible impurities. |
| **Interactions and Boundaries** | The nanostructures are separate and not fused together, with clear boundaries between each particle, indicating no significant interaction or aggregation. There are no different structures or phases visible; all particles appear identical and are evenly spaced. |
| **External Environment** | The image doesn't provide evidence of the nanomaterial interacting with its surrounding environment or matrix; it's a close-up with a focus solely on the nanoparticles. There are no other structures or objects visible that are not part of the nanomaterials themselves. The background is consistent with what you'd expect in electron microscopy - featureless, providing contrast to the particles being studied. |
| **Image Technique and Modifications** | The image was captured using Scanning Electron Microscopy (SEM). There's no apparent post-processing or modifications like false coloring or 3D rendering; it appears to be a standard grayscale SEM image. |
| **Functional Features** | No functional features such as active sites or regions with distinct properties are visible in the image. |
| **Context and Application** | The nanomaterial's application is unclear from the image. Such nanoparticles may serve in medicine, electronics, catalysis, or materials science. This looks like an experimental sample, not a simulation. |

1163

Table 19: The table presents an instruction-following dataset composed of image-question-answer triples generated using GPT-4 Turbo with vision in response to open-ended natural language prompts about a microscopic image of powder nanomaterials. In summary, this GPT-assisted vision-language dataset aids in the tuning of smaller models for the analysis and interpretation of complex microscopic images in the context of open-ended VQA tasks.

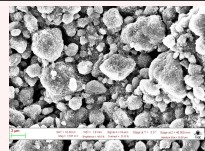

**Based on the provided image of powder nanomaterial, please answer the following questions.**

| Query prompt | GPT-4 Turbo with vision output |
|---|---|
| **Basics** | The image is a scanning electron microscope (SEM) image showing a powder nanomaterial. The scale bar indicates 2 micrometers (μm), meaning each unit on the scale bar represents 2 $\mu$m in the sample. |
| **Morphology and Structure** | The nanomaterials in the image exhibit an irregular, clumped morphology, with no distinct layers, phases, or domains readily apparent from this view. The particles do not appear uniform in size and shape; there is a variety of sizes and agglomerations, which is common in powdered nanomaterials due to their high surface energy leading to aggregation. |
| **Size and Distribution** | The individual nanostructures vary in size, but due to the scale bar indicating 2 micrometers, we can estimate that the particles range from sub-micrometer to a couple of micrometers in diameter. The distribution of nanomaterials throughout the image is clustered and random rather than evenly spaced, which is typical for powdered materials in a SEM image. There is evidence of aggregation or bundling, as many particles appear to be clumped together, forming larger aggregates rather than existing as isolated, individual particles. |
| **Surface Characteristics** | The nanomaterials in the image have a rough and textured surface, which is typical for powdered nanomaterials as seen under high magnification. There are no clearly visible defects, pores, or impurities that can be distinctly identified from the image provided. However, the granularity and irregularity of the surfaces suggest a high surface area, which is a common characteristic of nanomaterials. The roughness could contribute to potential sites for chemical activity or adsorption. |
| **Composition and Elements** | The SEM image shows uniform grayscale intensity, suggesting a homogeneous composition without visible evidence of different elements or compounds. There are no specific labels or markers indicating the presence of particular elements or compounds in the image. |
| **Interactions and Boundaries** | Individual nanostructures in the image appear to be touching and fused, forming clusters with no clear boundaries between them, indicating a degree of agglomeration. There are no visible distinct boundaries between different structures or phases. |
| **External Environment** | The SEM image shows only the nanomaterial tip, with no evidence of interaction with or immersion in an external environment or matrix. No other structures or materials are visible. |
| **Image Technique and Modifications** | The image was captured using Scanning Electron Microscopy (SEM). It appears to be a standard SEM output without false coloring or 3D rendering. SEM images are grayscale, and there are no signs of post-processing. |
| **Functional Features** | The SEM image does not visibly show any specific functional features like active sites. |
| **Context and Application** | The intended application of the nanomaterial in the SEM image cannot be determined from the image alone. It is an experimental sample, not a theoretical or simulation-based representation. |

Table 20: The table presents an instruction-following dataset composed of image-question-answer triples generated using GPT-4 Turbo with vision in response to open-ended natural language prompts about a microscopic image of MEMS devices and electrodes nanomaterials. In summary, this GPT-assisted vision-language dataset aids in the tuning of smaller models for the analysis of complex microscopic images in the context of open-ended VQA tasks.

**Based on the provided image of MEMS devices and electrodes nanomaterial, please answer the following questions.**

| Query prompt | GPT-4 Turbo with vision output |
| --- | --- |
| **Basics** | The image shows nanoscale electrodes or conductive paths, possibly part of a microelectromechanical system (MEMS). The scale bar indicates that each unit of measurement represents 2 micrometers. |
| **Morphology and Structure** | The nanomaterials in the image exhibit a linear, elongated morphology, likely representing nanowires or nanorods. There appear to be distinct parallel layers or strips, which could be separate phases or material components. The uniformity in the spacing between the layers suggests a controlled fabrication process. The nanomaterials themselves seem to have a consistent width, indicating a uniform size and shape across the visible structures. |
| **Size and Distribution** | The nanostructures in the image appear as straight, parallel lines with a consistent width less than 2 micrometers. They are evenly spaced and aligned, indicating a controlled distribution typical of fabricated nano-materials. There is no evidence of aggregation or bundling, supporting the idea of deliberate design often seen in MEMS devices. |
| **Surface Characteristics** | The nanomaterial's surface appears relatively smooth, with minor varia-tions that could be indicative of texture or grain boundaries. There are some visible defects and impurities, such as particles or irregularities on the surface between the parallel lines. |
| **Composition and Elements** | The monochromatic SEM image shows uniform brightness along the nanomaterial, suggesting homogeneous composition without visible variation. No labels indicate specific elements or compounds. |
| **Interactions and Boundaries** | The nanostructures are separate and do not appear to be touching or fused, indicating they are likely isolated conductive paths or electrodes. There are clear boundaries between the structures, as evidenced by the distinct, parallel lines that separate them. |
| **External Environment** | There is no evidence of interaction with a surrounding environment or matrix in the image. All visible features appear to be part of the nanomaterial system, with no other distinct objects present. |
| **Image Technique and Modifica-tions** | The imaging technique used is Scanning Electron Microscopy (SEM), as indicated by the details in the image. There's no evidence of post-processing modifications like false coloring or 3D rendering; the image appears to be a standard grayscale SEM image. |
| **Functional Features** | The image, being a static SEM representation, does not capture dynamic processes. As for functional features, the parallel linear structures likely represent active regions, such as conductive paths in a MEMS device. However, specific active sites or regions with distinct properties are not explicitly visible in this image. |
| **Context and Application** | SEM image reveals nanomaterial structure suited for electronics/MEMS (electrodes/conductors). The image is a real SEM photograph, not a theoretical or simulation-based representation. |

## 4.1 Additional datasets and Experimental results

To bolster the robustness and generalizability of our framework, we conducted evaluations using a diverse range of open-source benchmark datasets. These datasets are relevant to our research domain and encompass a broad spectrum of applications. This comprehensive evaluation strategy not only validated the efficacy of our framework but also demonstrated its adaptability to a wider range of datasets, extending beyond the SEM dataset[Aversa et al., 2018].

**NEU-SDD**

To thoroughly evaluate the effectiveness of our proposed method, specifically for open-ended VQA tasks involving multiple defect categories, we utilized the NEU-SDD dataset([Deshpande et al., 2020]) [1]. This dataset comprises an extensive collection of 1,800 electron microscopy images illustrating surface defects on hot-rolled steel plates. The NEU-SDD dataset enabled us to evaluate our framework's ability to comprehend complex visual information and provide insightful answers to questions about the surface defects. Each defect category in the NEU-SDD dataset is represented by 300 images, with each image having a resolution of 200×200 pixels. The dataset is categorized into six distinct types of defects, with 300 representative micrographs for each category. These categories encompass a diverse range of surface imperfections, including pitted surfaces, scratches, rolled-in scale, crazing, patches, and inclusion defects. Notably, each image in the dataset features only one type of defect. Figure 9 provides illustrative images from each category. In summary, the NEU-SDD dataset represents a valuable resource for the development and evaluation of surface defect-based VQA algorithms. Its diverse range of defects, and high-quality images make it a challenging and realistic benchmark for this task.

**CMI**

The CMI dataset[2], meticulously curated by corrosion experts, comprises 600 high-resolution electron micrographs that vividly capture the deterioration of corroded panels. These meticulously labeled images adhere to the ASTM-D1654 standards and feature individual scores ranging from 5 to 9, corresponding to 120 unique micrographs each. Each micrograph has a spatial resolution of $512 \times 512$ pixels, providing a granular view of the corrosion damage. Figure 10 showcases representative images from each score-based category. We conducted experimental studies to evaluate the effectiveness of our proposed technique for both multi-category classification and open-ended VQA tasks.

**KTH-Tips**

The KTH-TIPS[3] dataset, which serves as a cornerstone in texture analysis, comprises an extensive collection of 810 electron micrographs. Each of these images has been meticulously categorized into one of ten distinct material classes. These high-resolution images, each measuring $200 \times 200$ pixels, capture a diverse range of materials under varying lighting conditions, orientations, and scales. The comprehensive collection encompasses textures such as sponge, orange peel, styrofoam, cotton, cracker, linen, crust, sandpaper, aluminum foil, and corduroy. The representative images from each material class can be seen in Figure 11. To evaluate the effectiveness of our proposed method in multi-category texture-based classification and open-ended visual question answering (VQA) tasks, we conducted comprehensive experiments.

**Additional Information**

A common misconception is that GPT-4 Turbo with Vision can handle all tasks, from image classification to visual question answering (VQA), with a one-size-fits-all prompt. In reality, each task requires a carefully designed prompt specific to the dataset, leveraging our understanding of the model's capabilities. Diverse prompting strategies are essential in AI, not just beneficial. By tailoring prompts to individual needs, we unlock the full potential of advanced AI models and ensure generation of high-quality, instruction-following datasets. We leverage custom prompts tailored to each specific additional datasets. This allows us to generate instruction-following datasets focused on the material categories present in the input microscopy images. Subsequently, smaller models trained on this generated data can learn human intent from larger teacher models, ultimately achieving state-of-the-art performance on downstream tasks. To evaluate the effectiveness of the `MAEMI` framework, we conducted a comprehensive performance comparison with existing SOTA models across various tasks. Specifically for multi-class classification tasks, Table 21 presents classification accuracy results, demonstrating `MAEMI`'s performance relative to baseline models. In the domain of open-ended VQA, Table 22 showcases `MAEMI`'s performance, providing a detailed comparison with alternative approaches. To further illustrate `MAEMI`'s capabilities in open-ended VQA, Tables 30, 31, and 32 offer concrete examples presenting images, corresponding questions, and the generated answers. These tables go beyond textual comparison by incorporating performance evaluation metrics such as BLEU-2, ROUGE-L, and METEOR, ensuring a quantitative assessment. Additionally, Tables 23 - 29 present samples from the instruction-tuning Q&A pairs dataset, generated by GPT-4 Turbo with Vision. This dataset plays a crucial role in the training process of smaller models.

---

[1]Datasource: http://faculty.neu.edu.cn/yunhyan/NEU_surface_defect_database.html

[2]https://arl.wpi.edu/corrosion_dataset

[3]https://www.csc.kth.se/cvap/databases/kth-tips/index.html

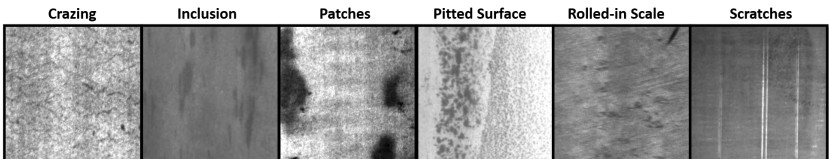

Figure 9: The figure displays a curated collection of electron microscopy images from the NEU-SDD dataset, also known as the NEU Surface Defect Database. This specialized dataset is primarily used for detecting and classifying surface defects on steel. It contains images representing six different types of steel surface defects found on hot-rolled steel strips: *pitted surfaces, scratches, rolled-in scale, crazing, patches, and inclusion defects*. The database plays a crucial role in developing frameworks for quality control in manufacturing and automated inspection systems by providing a diverse range of defect types and images for comprehensive testing and evaluation.

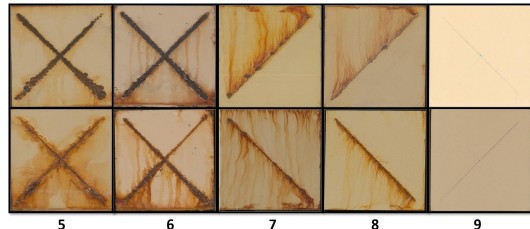

Figure 10: The figure shows a curated collection of electron micrographs from the CMI dataset, which have been methodically categorized based on the ASTM-D1654 standards. It features corrosion severity scores from 5 to 9, suggesting a scale that measures the progression of corrosion damage on the material panels. With scores ranging from 5 to 9 indicating a progression from moderate to less severe corrosion. The CMI dataset includes 600 images of material panels undergoing different levels of corrosion, each evaluated and confirmed by experts through standardized laboratory testing.

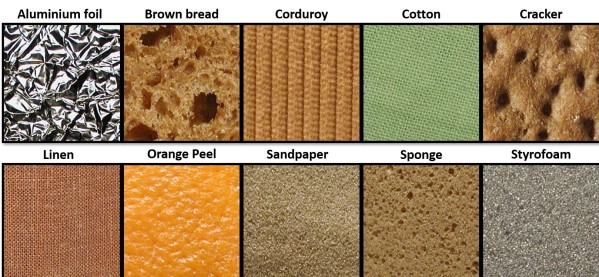

Figure 11: The figure shows a curated selection of electron micrographs from the KTH-TIPS texture dataset showcasing the ten diverse material classes, including *sponge, orange peel, styrofoam, cotton, cracker, linen, crust, sandpaper, aluminum foil, and corduroy*.

|  | Algorithms | NEU-SDD | CMI | KTH-TIPS |
|---|---|---|---|---|
| Baselines | ResNet | 0.906 | 0.928 | 0.941 |
| | GoogleNet | 0.936 | 0.928 | 0.929 |
| | SqueezeNet | 0.955 | 0.943 | 0.963 |
| | VanillaViT | 0.962 | 0.968 | 0.972 |
| | **MAEMI** | **0.985** | **0.983** | **0.991** |

Table 21: The table compares the multi-category classification performance of the proposed framework against established benchmarks across datasets.

Table 22: The table shows `MAEMI` framework excels on open-ended VQA task across benchmark datasets with their corresponding scores in several evaluation metrics.

| Method | BLEU-2 (↑) | BLEU-4 (↑) | ROUGE-1 (↑) | ROUGE-2 (↑) | ROUGE-L (↑) | METEOR (↑) |
|---|---|---|---|---|---|---|
| InstructBLIP[Dai et al., ] | 0.822 | 0.666 | 0.933 | 0.832 | 0.895 | 0.955 |
| LLaVA[Liu et al., 2023] | 0.814 | 0.664 | 0.956 | 0.841 | 0.903 | 0.951 |
| MiniGPT-4[Zhu et al., 2023] | 0.855 | 0.687 | 0.966 | 0.828 | 0.901 | 0.982 |
| **MAEMI** | **0.908** | **0.823** | **0.984** | **0.889** | **0.942** | **0.992** |

Table 23: The table presents an instruction-following dataset composed of image-question-answer triples generated using GPT-4 Turbo with vision in response to open-ended natural language prompts about a microscopic image of corrosion category 5, according to ASTM-D1654 standards. In summary, this GPT-assisted vision-language dataset aids in tuning smaller models to interpret complex microscopic images in the context of open-ended VQA tasks.

**Please answer the following questions about the provided image of corrosion category 5, classified according to ASTM-D1654 standards.**

| Query prompt | GPT-4 Turbo with vision output |
| --- | --- |
| What is the primary purpose of the image? | The image is used for research on material corrosion, specifically to classify the extent of damage according to standardized corrosion severity levels in a Visual Question Answering system. It's a part of a dataset for training machine learning models in corrosion detection. |
| How is the severity of corrosion visually represented in this rating? | The severity of corrosion in the images is determined by the extensiveness of the corroded area- in terms of size, depth, color changes, and presence of rust or pitting. The lower ASTM-D1654 rating scores like a 5 here indicate more severe corrosion, with larger and more numerous defects visible on the metal surface. |
| What is the dominant feature in the image? | The image's dominant feature is a large X-shaped mark, suggesting severe corrosion or a deliberate assessment marking in the corrosion dataset. Its size, shape, and contrasting color against the surface indicate a lower corrosion severity rating, implying significant damage or deterioration. |
| Describe the color and texture of the X-shape. | The X-shape in the image is dark, suggesting heavy oxidation or degradation. The texture is rough and uneven, hinting at flaking or pitting typical of corroded metal. The color contrasts with the lighter background, possibly indicating early corrosion stages or intact coatings. This is a typical appearance of a surface that has experienced significant corrosion. |
| What's notable about the background surrounding the X-shape? | The background has a mottled appearance with color variations from lighter to darker shades, resembling rust and wear from corrosion. The uneven surface shows pitting and discoloration, suggesting the onset of corrosion. This context indicates the X-shape is part of a larger corroded area being studied for material degradation. |
| Are there visible boundaries or divisions between the X-shape and the background? | The distinct X-shape boundaries suggest a clear division from the background, possibly due to deeper corrosion setting it apart or differences in image processing or metal treatment in that area. |
| Is there a pattern to the corrosion emanating from the X-shape? | Generally, corrosion may spread outward from points of origin, but the exact pattern would depend on various factors including the material and environmental conditions. |
| How do the dark speckles on the background contribute to the overall appearance of corrosion? | Dark speckles on a corroded surface indicate areas where the material has started to pit or the protective layer has been compromised, allowing oxidation. Their varying size and density give insights into the severity and progression of corrosion. They create visual texture reflecting the irregular nature of corrosion across the surface. |
| What does a rating of 5 signify in terms of material integrity and longevity? | A corrosion rating of 5 signifies severe degradation that can compromise integrity and longevity, requiring attention, maintenance, or replacement to ensure safety and functionality. |

Table 24: The table presents an instruction-following dataset composed of image-question-answer triples generated using GPT-4 Turbo with vision in response to open-ended natural language prompts about a microscopic image of corrosion category 9, according to ASTM-D1654 standards. In summary, this GPT-assisted vision-language dataset aids in tuning smaller models to analyze and interpret complex microscopic images in the context of open-ended VQA tasks..

**Please answer the following questions about the provided image of corrosion category 9, classified according to ASTM-D1654 standards.**

| Query prompt | GPT-4 Turbo with vision output |
| --- | --- |
| What is the primary intent of the image? | The primary intent of the image you've provided, which belongs to the corrosion 9 category from the CMI dataset, is to serve as a detailed electron micrograph for the study of corrosion. It appears to capture the advanced state of deterioration on metal panels, providing a high-resolution view to analyze and understand the patterns and progression of corrosion. This kind of image is typically used by researchers and professionals in the field of material science and engineering to assess the condition of metal structures, to develop better corrosion-resistant materials, and to test the effectiveness of anti-corrosion treatments. The image's adherence to ASTM-D1654 standards indicates that it has been categorized and scored based on a standardized measure of evaluating the severity of corrosion, which is crucial for consistent scientific analysis and research. |
| Can you describe the overall color scheme of the image? | The image displays a predominantly beige or tan background with a central feature that appears as a rust or brownish streak, indicative of corrosion. This streak has variations in intensity, with some areas appearing darker, suggesting different stages or concentrations of corrosion. |
| What is the main corroded feature present in the image? | The main corroded feature in the image is a long, linear streak of corrosion running across the field of view. This streak is characterized by its brownish, rust-like color, which contrasts with the lighter beige background. It likely represents a significant area of metal degradation, with variations in color intensity indicating different levels or depths of corrosion damage. |
| How does the corrosion manifest around this scribed line? | The corrosion shows as a rust-colored line where the coating was likely removed, allowing corrosion to start. Around this line, the corrosion spreads outward, creating a gradient where the rust color fades away from the central line. This indicates severe corrosion near the scribed line, with less impact on the surrounding metal. |
| Given the context of corrosion ratings, how does a rating of 9 compare to other ratings in terms of corrosion severity? | A rating of 9 in the context of corrosion severity, especially within a scale that adheres to ASTM-D1654 standards, typically indicates minimal corrosion, with the metal surface remaining mostly intact. Lower ratings would denote more severe corrosion damage. |
| Is the corrosion consistent throughout the scribed line? | The corrosion along the scribed line is not uniform; it varies in intensity, suggesting different levels of corrosion severity along the line. |
| How does the background of the image appear in comparison to the corroded area? | The background of the image is a uniform beige or tan color, which appears smooth and undamaged, contrasting with the corroded area that shows a brownish, textured streak indicative of corrosion damage. |
| If one were to run their fingers across the image, what differences in texture might they expect between the corroded scribe and the background? | Touching the corroded panel in the image would reveal roughness and unevenness along the corroded line, contrasting with the smoother, intact background. The corroded area feels irregular and pitted, while the surrounding area is smoother. However, this texture is only visually perceptible in the image, not tactile. |

Table 25: The table presents an instruction-following dataset composed of image-question-answer triples generated using GPT-4 Turbo with vision in response to open-ended natural language prompts about a microscopic image of the aluminium foil. In summary, this GPT-assisted vision-language dataset aids in tuning smaller models to analyze and interpret complex microscopic images in the context of open-ended VQA tasks.

**Referring to the provided image of aluminum foil, please provide answers to the questions below.**

| Query prompt | GPT-4 Turbo with vision output |
|---|---|
| What is the primary material depicted in the image? | The primary material depicted in the image you provided appears to be aluminum foil. Aluminum foil is typically characterized by its shiny surface and its ability to crumple and fold easily, creating a range of highlights and shadows that give it a distinctive textured appearance. The image shows these characteristics, suggesting that it is indeed aluminum foil. |
| How is the surface texture of the material? | The surface texture of the aluminum foil in the image is highly irregular and crinkled, characterized by sharp folds and creases that reflect light differently. This creates a complex pattern of highlights and shadows. The crumpled nature results in a chaotic, random texture typical of manipulated foil. The texture varies with the degree of crumpling and lighting angle, contributing to the foil's unique and detailed appearance. |
| What color predominates in the image? | The image predominantly displays shades of gray, characteristic of the metallic sheen of aluminum foil. The varying brightness within those shades is due to the crinkled texture and how it reflects light. |
| Are there any reflections or shines visible on the material? | Yes, there are multiple reflections and shines visible on the material, which are indicative of aluminum foil's reflective metallic surface. The crinkles and folds create varying degrees of brightness and contrast, emphasizing the shiny quality of the foil. |
| How would you describe the overall pattern on the foil? | The overall pattern on the aluminum foil is highly irregular and chaotic, with no uniformity or repeating motifs. It consists of numerous random creases and folds that create a complex and intricate texture. The pattern is formed by the way the foil has been crumpled, resulting in a disordered array of reflective angles and surfaces. |
| Is the foil flat or does it appear to have depth and volume? | The foil appears to have depth and volume due to the pronounced creases and folds, which create a three-dimensional effect, the crumpling gives it a textured appearance that suggests a sense of depth. |
| Does the foil cover the entirety of the image? | The foil appears to cover the entirety of the image with no other distinct elements or objects present. |
| How would you describe the lighting on the foil? | The lighting on the foil seems to be diffuse, with no harsh or direct reflections, suggesting a soft, indirect light source. This type of lighting enhances the visibility of the texture by casting gentle shadows within the creases without creating overpowering glare. |
| Is there any uniformity in the crinkles and folds, or are they random? | The crinkles and folds in the aluminum foil appear to be random and without any discernible pattern or uniformity. Each crease and fold is unique, contributing to the overall complex texture of the material. |
| Is the background distinguishable, or does the foil cover it completely? | The foil covers the background completely, making it indistinguishable. There are no visible elements or details beyond the foil itself in the image. |

Table 26: The table presents an instruction-following dataset composed of image-question-answer triples generated using GPT-4 Turbo with vision in response to open-ended natural language prompts about a microscopic image of the corduroy fabric. In summary, this GPT-assisted vision-language dataset aids in tuning smaller models to analyze and interpret complex microscopic images in the context of open-ended VQA tasks.

**Referring to the provided image of corduroy fabric, please provide answers to the questions below.**

| Query prompt | GPT-4 Turbo with vision output |
|---|---|
| What type of material is depicted in the image? | The image you've provided appears to be a texture that is consistent with the material class of corduroy. Corduroy is characterized by its distinct pattern of raised ridges, or wales, which run in parallel lines. The texture in the image exhibits these hallmark ridges, suggesting that it indeed represents a corduroy material. |
| What color does the corduroy fabric primarily exhibit? | The corduroy fabric in the image primarily exhibits a brown color. |
| Describe the texture and pattern noticeable on the corduroy | The texture of the corduroy in the image is characterized by its distinctive ridges or wales, which are raised parallel lines that run along the length of the fabric. These wales give corduroy its unique texture that is both visual and tactile. The pattern is repetitive and consistent, with each wale typically having a uniform width and height, creating a corded look that is typical of this type of fabric. The valleys between the wales appear as linear depressions that alternate with the raised wales, contributing to the overall texture of the material. The texture also suggests a soft and somewhat velvety feel, which is common with corduroy fabrics. |
| Are these raised lines uniformly distributed throughout the fabric? | Yes, in the image of the corduroy fabric, the raised lines, or wales, are uniformly distributed, giving it a consistent texture and appearance. The regular spacing and similar width of each wale contribute to the material's uniform look. |
| How does light interact with the fabric's texture? | In the image, light interacts with the fabric's texture, creating highlights and shadows along the corduroy's wales. The raised ridges catch light, enhancing brightness and sheen on the peaks, while the valleys remain shadowed, accentuating the fabric's depth and texture. This interplay enhances the corduroy's three-dimensional look and emphasizes its tactile quality. The light's directionality also affects texture visibility; angled light increases the contrast between the wales and grooves, making the texture more pronounced. |
| Describe the spacing between the ribs. | The spacing between the ribs in the corduroy is uniform and even, typical of the regular pattern associated with this material. |
| How does the fabric's texture vary between the raised lines and the gaps in between? | The fabric's texture varies between the raised lines (wales) and the gaps (channels) in that the wales are more pronounced and may feel thicker or plush, while the channels are recessed, creating a softer and smoother feel in comparison. |
| Are there any visible imperfections or signs of wear on the corduroy? | The image of the corduroy does not exhibit any visible imperfections or signs of wear; the wales appear consistent and intact, indicating the fabric is likely in good condition. |
| From which angle or perspective is the fabric being observed? | The fabric is being observed from a direct, overhead angle, looking straight down onto the surface, allowing for a clear view of the texture and pattern of the corduroy. |
| Is the image a full representation of the fabric or just a section? | The image seems to represent just a section of the corduroy fabric, not the full extent of the material. It provides a close-up view that focuses on the texture and pattern detail. |

Table 27: The table presents an instruction-following dataset composed of image-question-answer triples generated using GPT-4 Turbo with vision in response to open-ended natural language prompts about a microscopic image of the linen fabric. In summary, this GPT-assisted vision-language dataset aids in tuning smaller models to analyze and interpret complex microscopic images in the context of open-ended VQA tasks.

**Referring to the provided image of the linen fabric, please provide answers to the questions below.**

| Query prompt | GPT-4 Turbo with vision output |
|---|---|
| What is the primary material displayed in the image? | The image is a representation of a linen texture. Linen is known for its distinctive, slightly irregular weave and natural fiber look, which is visible in the image provided. |
| Can you identify the primary color of the linen fabric in the image? | The primary color of the linen fabric in the uploaded image appears to be a beige or light tan color. This color is often associated with natural linen materials. |
| Describe the texture or pattern visible on the linen? | The linen's texture in the image shows a distinct, woven pattern typical of linen fabrics. The weave is tight and regular, with slight thread thickness variation, giving it a natural, irregular appearance. The pattern features a crisscross arrangement, creating a grid-like structure. The surface has a matte finish, suggesting a soft, natural feel typical of linen textiles. |
| How would you describe the spacing between the woven threads? | The spacing between the woven threads in the image appears to be quite close, but with enough separation to make individual threads distinguishable. The threads are woven in a consistent pattern, indicative of a tightly woven fabric, yet they are not so tightly bound that they lose the textured appearance distinctive to linen. This type of spacing is typical for quality linen, allowing for durability while retaining breathability. |
| Does the linen appear soft or rigid based on the visible texture? | Based on the visible texture in the image, the linen appears to have a soft quality. The slight irregularities in the thread thickness and the natural weave pattern suggest flexibility, which is characteristic of linen fabrics. Despite its structured and tight weave, linen is known for its natural softness and the image does not suggest any stiffness that would indicate rigidity. |
| Are there any observable sheen or reflective properties on the linen? | From the image provided, the linen does not exhibit a significant sheen or reflective properties. The surface appears matte, with the natural, slightly uneven texture of the weave absorbing light rather than reflecting it. This lack of luster is typical for linen, which is known for its more subdued, earthy aesthetic rather than a glossy or shiny appearance. |
| How consistent is the texture throughout the displayed section? | The texture of the linen in the image is consistent, with a uniform weaving pattern indicating high-quality textile production. Subtle natural variations in thread thickness, characteristic of linen, don't disrupt the fabric's overall uniformity. This consistent weave suggests a smooth, even feel, desirable in linen fabrics. |
| Based on the visible attributes, how would you describe the tactile sensation of the fabric? | The linen fabric in the image would likely offer a tactile experience that is characteristically textured and crisp to the touch, with a natural, slightly irregular feel indicative of its organic origin. The tight yet breathable weave suggests a fabric that is both durable and comfortable, providing a sensation of both softness and structured quality. |
| Are there any discernible imperfections or unique characteristics in the fabric? | Based on the image, the fabric is uniform, free from snags, pulls, or discoloration. The natural variation in thread thickness, characteristic of linen, provides a distinct, organic texture, enhancing its aesthetic and tactile appeal. Overall, the fabric appears high quality, embodying the nuanced details typical of woven linen. |

# 7 NEU-SDD

Table 28: The table presents an instruction-following dataset composed of image-question-answer triples generated using GPT-4 Turbo with vision in response to open-ended natural language prompts about a microscopic image of the scratches category of surface defects. In summary, this GPT-assisted vision-language dataset aids in tuning smaller models to analyze and interpret complex microscopic images in the context of open-ended VQA tasks.

**Based on the provided image from the scratches category, please respond to the following questions.**

| Query prompt | GPT-4 Turbo with vision output |
| --- | --- |
| What category does this texture belong to? | The category of the texture in the provided image belongs to the scratches category. Scratches are characterized by linear markings or indentations on the surface, which can vary in depth and width. These defects are typically a result of mechanical damage where an object has come into contact with the surface with enough force to mar it. In the context of the NEU-SDD dataset, this image would be used to train algorithms to recognize and categorize such imperfections on hot-rolled steel plates. |
| Describe the central feature in the image? | The central feature in the image is a linear, vertical scratch mark, characterized by a distinct, bright line against a darker background, indicative of a surface defect on the material. |
| How would you describe the contrast between the scratch and the surrounding texture? | The contrast between the scratch and the surrounding texture is quite pronounced. The scratch appears as a bright, linear streak against a uniformly darker background, which indicates a significant difference in reflectivity or texture between the scratch and the rest of the surface. This high-contrast feature makes the scratch readily identifiable. |
| What color tones dominate the image? | The image is grayscale, so it is dominated by shades of gray. There are no color tones in a grayscale image, only variations from black to white. |
| Can you define the orientation of the scratch? | The orientation of the scratch is vertical, extending from the top to the bottom of the image. |
| How would you describe the edges of the scratch? | The edges of the scratch appear relatively straight and parallel, suggesting a singular, consistent force that created it. They are well-defined, indicating a clear demarcation between the scratch itself and the surrounding unaffected material. |
| Is the scratch in the center or towards the side of the image? | The scratch is centrally located in the image, running vertically down the middle of the frame.. |
| Are there multiple scratches or just a single prominent one? | Based on the image provided, there is a single prominent scratch visible running vertically down the center. |
| Is the rest of the image's texture smooth or grainy? | The rest of the image's texture appears grainy, with a speckled pattern of lighter and darker gray areas indicative of the microscopic surface texture of the material. |
| How does the brightness of the scratch compare to the overall image? | The scratch is significantly brighter than the overall image. It stands out as a distinct, lighter feature against the predominantly darker, grainy background of the rest of the image. |
| If you were to touch the scratch, would it feel raised or recessed? | Based on the visual characteristics of the scratch in the image, it would likely feel recessed. The brightness of the scratch, in contrast to the darker surrounding area, suggests a groove or indentation in the material's surface. |

Table 29: The table presents an instruction-following dataset composed of image-question-answer triples generated using GPT-4 Turbo with vision in response to open-ended natural language prompts about a microscopic image of the patches category of surface defects. In summary, this GPT-assisted vision-language dataset aids in tuning smaller models to analyze and interpret complex microscopic images in the context of open-ended VQA tasks.

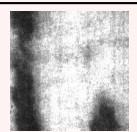

**Based on the provided image from the patches category, please respond to the following questions.**

| Query prompt | GPT-4 Turbo with vision output |
| --- | --- |
| What category does this texture belong to? | The texture in the image belongs to the patches category of surface defects. |
| Describe the dominant features in the image? | The image shows a grayscale texture with varied patterns of light and dark areas. Dominant features include irregularly shaped darker patches distributed across a lighter background, suggesting an uneven surface typical of the "patches" defect category in steel. These patches appear as areas of concentration or clusters, contrasting with the lighter grayscale background, and may indicate surface inconsistencies or repairs. |
| How would you characterize the contrast between the patches and the surrounding area? | The contrast between the patches and the surrounding area in the image is quite pronounced. The patches appear as distinct, darker areas against a lighter background, indicating a significant difference in tone. This stark contrast helps to highlight the irregularities and the texture differences between the defective patches and the rest of the surface. |
| What color tones are prevalent in the image? | The image is in grayscale, showing black, white, and gray tones. It lacks color, displaying intensity variations from black (low intensity or absence of material) to white (high intensity or presence of material), with gray shades indicating levels in between. The darker gray patches suggest lower intensity areas compared to the lighter gray surrounding surface. |
| Describe the shape and arrangement of the patches? | The patches in the image are irregularly shaped and vary in size and form, distributed randomly across the surface. Some cluster together, while others are isolated, with jagged, poorly-defined edges blending into the lighter background. This irregularity in shape and arrangement typifies material surface defects. |
| How does the texture within the patches compare to the rest of the image? | Within the patches, the texture appears to be denser and more chaotic compared to the rest of the image. The granularity is more pronounced, with a rougher, less uniform appearance. In contrast, the surrounding areas exhibit a smoother and more consistent texture with lighter shades of gray, suggesting a less disturbed surface. The patches stand out due to their distinct, rugged texture against the relatively uniform background. |
| Would you say the patches are centralized or more dispersed? | The patches appear to be dispersed throughout the image rather than centralized. They are spread across the surface without a clear concentration in any particular central area, indicating a random distribution of the surface defects. |
| Do the patches have well-defined boundaries? | The patches do not have well-defined boundaries; their edges are irregular and blend into the surrounding area. |
| Describe the granularity of the background texture? | The background texture in the image is relatively smooth and uniform, with fine granularity. This contrasts with the rougher texture of the patches, making the overall surface appear heterogeneous. |
| In a real-world scenario, how might such patches form on materials? | Patches on materials can form due to corrosion, wear and tear, manufacturing flaws, inconsistent repairs or treatments, and chemical exposure, often indicating areas of weakness. |
| Does the image lean more towards being bright, dark, or neutral overall? | The image is overall neutral in tone, with a balance between light and dark areas, neither leaning distinctly towards being bright nor dark. |

Table 30: The table displays a collection of electron microscope images that depict metal corrosion, accompanied by their accurate labels. Additionally, it includes machine-generated descriptions for each image, which are obtained from an open-ended VQA task that examines contributing factors. The effectiveness of these generated descriptions is assessed by comparing their similarity to the actual labels, using BLEU-2, ROUGE-L, and METEOR evaluation metrics.

| Image | Ground Truth | Answers | BLEU-2/ ROGUE-L/ METEOR |
|---|---|---|---|
|  | The corrosion on this metal could be influenced by environmental factors (moisture, temperature, exposure to chemicals), the metal's composition, protective coating failures, and possibly physical damage that compromised the surface. | The corrosion on this metal could be due to environmental factors (moisture, temperature, chemical exposure), the metal's composition, protective coating failures, and possibly physical damage compromising the surface | 0.689 0.833 0.802 |
|  | The corrosion shown may be due to environmental elements, material defects, or inadequate corrosion protection measures. | The corrosion shown may be due to environmental factors, defects in materials, or inadequate corrosion protection measures. | 0.725 0.848 0.806 |
|  | Corrosion here appears influenced by direct exposure to elements, possibly exacerbated by an abrasive treatment that may have removed protective coatings. | Corrosion appears influenced by direct exposure to elements, possibly exacerbated by an abrasive method that may have removed some protective coatings. | 0.824 0.905 0.901 |
|  | The streaking patterns suggest water flow, so exposure to moisture and environmental conditions likely played roles in the corrosion process. | The streaking patterns indicate water flow, suggesting that exposure to moisture and environmental factors likely played roles in the corrosion process. | 0.725 0.829 0.886 |
|  | In this image, corrosion seems minimal, suggesting effective protective measures, possibly the trivalent chromium pretreatment, are in place. | In this image, corrosion appears minimal, indicating that effective protective measures, possibly including the trivalent chromium pretreatment, are in place. | 0.711 0.842 0.931 |

Table 31: This table showcases a selection of electron microscope images alongside their corresponding labels, and framework-generated descriptions on an open-ended VQA task delving into material properties like durability, degradation, and environmental impact. To gauge the effectiveness of the automatically generated descriptions for these images, we evaluated their similarity to the true labels using metrics such as BLEU-2, ROUGE-L, and METEOR.

| Image | Ground Truth | Answers | BLEU-2/ ROGUE-L/ METEOR |
|---|---|---|---|
|  | The material shown, likely aluminum foil, is durable and resistant to degradation but can have a significant environmental impact if not recycled properly. | The shown material, likely aluminum foil, is durable and resistant to degradation but can have a significant environmental impact if not recycled correctly. | 0.844 0.913 0.869 |
|  | The material appears to be bread, with low durability, quick biodegradation, and minimal environmental impact. | The material, which appears to be bread, possesses low durability, quick biodegradation, and minimal environmental impact. | 0.736 0.903 0.855 |
|  | The material appears to be carpet, which is moderately durable, degrades over years, and can be environmentally impactful if synthetic and not recycled. | The material, appearing to be carpet, possesses moderate durability, degrades over time, and can be environmentally significant if made of synthetic materials and not recycled. | 0.447 0.667 0.727 |
|  | This textile, possibly canvas or burlap, has high durability, slow degradation, and if natural, a low environmental impact. | This textile, which could be canvas or burlap, has high durability, degrades slowly, and has a low environmental impact if it is natural | 0.453 0.634 0.648 |
|  | The material, likely sponge or foam, is less durable, degrades variably, and can have a higher environmental impact if not biodegradable. | The material, presumably sponge or foam, is comparatively less durable, degrades in various ways, and potentially has a higher environmental impact if it lacks biodegradability. | 0.500 0.652 0.688 |
|  | The fabric, likely a natural fiber weave, is moderately durable, biodegradable, and has a low environmental impact when untreated. | The fabric, likely comprised of a natural fiber weave, is moderately durable, biodegrades effectively, and usually has a minimal environmental impact when untreated | 0.635 0.810 0.865 |
|  | The material, resembling an organic rind, has moderate durability, biodegradable properties, and a low environmental impact. | The substance, which appears to be an organic rind, exhibits moderate durability, has biodegradable qualities, and generally results in a low environmental impact. | 0.393 0.615 0.718 |
|  | The material, likely sandpaper, is designed for short-term use, degrades with wear, and has a moderate environmental impact depending on the backing material. | The substance, resembling sandpaper, is constructed for limited use, degrades with wear, and has a moderate environmental impact, varying with the backing. | 0.499 0.696 0.573 |
|  | The material, resembling a sponge, has low to moderate durability, variable degradation, and a potentially high environmental impact if synthetic. | The substance, which looks like a sponge, has low to moderate durability, degrades variably, and can have a high environmental impact if it is synthetic. | 0.474 0.615 0.708 |
|  | The material, likely glittery fabric or paper, has low to moderate durability, can degrade slowly, and often has a high environmental impact due to microplastic pollution. | The substance, possibly glittery fabric or paper, displays low to moderate durability, tends to degrade slowly, and often leads to a high environmental impact from microplastic pollution. | 0.609 0.696 0.731 |

Table 32: The table displays a selection of electron microscope images along with their corresponding labels and framework-generated descriptions. These descriptions are evaluated for their effectiveness in a open-ended VQA task that investigates defects and their identifying features. We assess the similarity between the automatically generated descriptions and the true labels using metrics such as BLEU-2, ROUGE-L, and METEOR.

| Image | Ground Truth | Answers | BLEU-2/ ROGUE-L/ METEOR |
|---|---|---|---|
|  | The image displays crazing, characterized by a network of fine, inter-linked cracks on the surface. | The picture shows crazing, marked by a fine, interconnected network of cracks on its surface | 0.378 0.600 0.721 |
|  | The image shows the defect known as inclusion, identifiable by darker areas or spots embedded within the material's matrix. | The image exhibits the defect known as inclusion, evident from the darker areas or spots within the material's matrix. | 0.695 0.800 0.820 |
|  | The image shows a defect known as patches, which are characterized by large, dark, and irregularly shaped areas on the material's surface. | The image depicts the defect referred to as patches, characterized by expansive, dark, and irregularly shaped regions on the surface. | 0.460 0.651 0.703 |
|  | The image shows a pitted surface defect, characterized by numerous small, shallow depressions scattered across the material's surface. | The image illustrates a pitted surface condition, with numerous small and shallow depressions distributed across the material's surface. | 0.524 0.737 0.643 |
|  | The image shows the defect known as rolled-in scale, indicated by dark streaks and spots embedded in the surface, typical of metalworking. | The image illustrates the defect known as rolled-in scale, recognizable by the dark streaks and embedded spots on its surface, often seen in metalworking. | 0.511 0.667 0.741 |
|  | The image shows a defect called scratches, identifiable by the long, thin, and straight lines running across the material's surface. | The image reveals scratches as a defect, recognizable by the long, slender, and straight paths etched across the material's face. | 0.417 0.619 0.524 |