# OpenReview forum: "Autonomous AI Assistant for Semiconductor Electron Micrograph Analysis: Instruction-Tuning Small-Scale Language-and-Vision Assistant for Enterprise Adoption in Low-Resource Settings"
_ijcai.org/IJCAI/2024/Workshop/AI4Research — AI4Research 2024_

### Official Review · Reviewer_8nA4 · 2024-06-01
**This work distills semiconductor electron microscopy knowledge from GPT-4 Turbo with Vision to a small scale model.**

**Rating:** 7
**Confidence:** 4

**Review:**

This work presents MAEMI a small scale model that is specialized to semiconductor electron microscopy images. The model has been trained on a multimodal instruction data collected using GPT4-Turbo with Vision.

Strengths:
The proposed model shows strong performance on semiconductor electron microscopy images. It outperforms similar sized models on various tasks such as VQA, classification and image captioning.

Weakness:
Details of evaluation datasets such as dataset size etc are missing. Also performance comparison against GPT-4 is missing. Since the dataset is generated using GPT-4, the performance of GPT-4 should indicate an upper bound. It is important to see how close to this bound the proposed model can get to. Also human evaluation of GPT-4 generated data has not been performed. So the quality of the dataset is unknown.

Suggestions:
The paper can be made much concise by removing  many technical details such as explanations of DyQLoRA-FA. This discussion is not central to the paper. More space should be given to discussion of data generation process.

---

### Official Review · Reviewer_gNed · 2024-06-04
**The paper introduces MAEMI, demonstrating strong results, improvements in organization and updating baselines are suggested.**

**Rating:** 6
**Confidence:** 4

**Review:**

This paper introduces MAEMI, a small-scale multimodal framework designed for the analysis of semiconductor electron microscopy images. It leverages vision-language instruction tuning to perform tasks such as image classification, captioning, and visual question answering (VQA) without relying on expensive, human-annotated datasets. The primary problem addressed by the study is the scarcity of high-quality, annotated datasets required for effective microscopic image analysis in semiconductor manufacturing. The authors propose a method that uses larger multimodal models to generate instruction-following datasets, which are then used to train smaller, more efficient models through knowledge distillation. The experiments seem solid, where the proposed method gets better results across both zero-shot and few-shot settings. The main issue of this paper is the writing.

Strengths:
The multimodal microscopy image analysis sounds novel to me.
Solid experiments.

Weaknesses:
The paper is not well-written. The section logic of this paper can be further improved. For example, Section 1 “Introduction” has 9 subsections, but Section 2 “Experiment” only has one section. Should better move some subsections from “Introduction” to “Experiment”, such as Section 1.9 “Evaluation Metrics”.
There are too many separate figures to illustrate the proposed framework, which can be combined and made in a more concise format (in my opinion). For example, tons of information are shared between Figure 3 and 4. This is really bad for the reader to understand.
All the baselines in the Tables are out-of-date — all of them are before 2021 — are there any other up-to-date baselines that can be applied to this task?

---

### Decision · Program_Chairs · 2024-06-03

Accept